# Increased mTOR activity and metabolic efficiency in mouse and human cells containing the African-centric tumor-predisposing p53 variant Pro47Ser

Keerthana Gnanapradeepan[1,2], Julia I-Ju Leu[3], Subhasree Basu[1], Thibaut Barnoud[1], Madeline Good[1], Joyce V Lee[4], William J Quinn[5], Che-Pei Kung[6], Rexford Ahima[7], Joseph A Baur[5], Kathryn E Wellen[4], Qin Liu[1], Zachary T Schug[1], Donna L George[3], Maureen E Murphy[1]*

[1]Program in Molecular and Cellular Oncogenesis, The Wistar Institute, Philadelphia, United States; [2]Graduate Group in Biochemistry and Molecular Biophysics, Perelman School of Medicine, University of Pennsylvania, Philadelphia, United States; [3]Department of Genetics, Perelman School of Medicine, University of Pennsylvania, Philadelphia, United States; [4]Department of Cancer Biology, Perelman School of Medicine, University of Pennsylvania, Philadelphia, United States; [5]Department of Physiology and Institute for Diabetes, Obesity, and Metabolism, Perelman School of Medicine, University of Pennsylvania, Philadelphia, United States; [6]Washington University in St. Louis, St Louis, United States; [7]Division of Endocrinology, Diabetes & Metabolism, Johns Hopkins University School of Medicine, Baltimore, United States

*For correspondence:
mmurphy@wistar.org

**Abstract** The Pro47Ser variant of p53 (S47) exists in African-descent populations and is associated with increased cancer risk in humans and mice. Due to impaired repression of the cystine importer *Slc7a11*, S47 cells show increased glutathione (GSH) accumulation compared to cells with wild-type p53. We show that mice containing the S47 variant display increased mTOR activity and oxidative metabolism, as well as larger size, improved metabolic efficiency, and signs of superior fitness. Mechanistically, we show that mTOR and its positive regulator Rheb display increased association in S47 cells; this is due to an altered redox state of GAPDH in S47 cells that inhibits its ability to bind and sequester Rheb. Compounds that decrease glutathione normalize GAPDH-Rheb complexes and mTOR activity in S47 cells. This study reveals a novel layer of regulation of mTOR by p53, and raises the possibility that this variant may have been selected for in early Africa.

## Introduction

The p53 tumor suppressor protein serves as a master regulator of the cellular response to intrinsic and extrinsic stress. Mutations in the *TP53* gene occur in more than 50% of human cancers, and this gene is well known as the most frequently mutated gene in cancer (*Hollstein et al., 1991*). p53 works to suppress uncontrolled cellular growth and proliferation through various pathways including apoptosis, senescence, cell cycle arrest, and ferroptosis (*Stockwell et al., 2017*; *Vousden and Prives, 2009*). More recently, a role for p53 in the control of metabolism has emerged. The metabolic functions of p53 include the regulation of mitochondrial function, autophagy, cellular redox

state, and the control of lipid and carbohydrate metabolism; for review see *Berkers et al., 2013*; *Gnanapradeepan et al., 2018*.

As an integral part of its control of metabolism, p53 negatively regulates the activity of mTOR (mammalian target of rapamycin), which is a master regulator of metabolism in the cell. mTOR is a serine-threonine protein kinase that is stimulated by mitogenic signals, and phosphorylates down-stream targets that in turn regulate protein synthesis and cell growth (*Ben-Sahra and Manning, 2017*). mTOR exists in two distinct signaling complexes: mTORC1 is primarily responsible for cell growth and protein synthesis, while mTORC2 plays roles in growth factor signaling, cytoskeletal control, and cell spreading (*Liu and Sabatini, 2020*). Not surprisingly, mTOR activity is frequently upre-gulated in a diverse range of cancers. p53 negatively regulates the mTOR pathway in part through transactivation of the target genes *PTEN*, *TSC2*, *PRKAB1* and *SESN1/SESN2* (*Budanov and Karin, 2008*; *Feng et al., 2005*). The regulation of mTOR by p53 is believed to couple the control of genome integrity with the decision to proliferate (*Hasty et al., 2013*).

*TP53* harbors several functionally impactful genetic variants or single-nucleotide polymorphisms (SNPs) (*Basu et al., 2018*; *Jennis et al., 2016*; *Kung et al., 2016*). A naturally occurring SNP in *TP53* exists at codon 47, encoding serine instead of a proline (Pro47Ser, rs1800371, G/A). This variant exists predominantly in African-descent populations, and occurs in roughly 1% of African Americans and 6% of Africans from sub-Saharan Africa (*Murphy et al., 2017*). The S47 variant is associated with increased risk for pre-menopausal breast cancer in African American women (*Murphy et al., 2017*). In a mouse model, the S47 mouse develops markedly increased incidence of spontaneous cancer, particularly hepatocellular carcinoma (*Jennis et al., 2016*). This variant is likewise defective in the regulation of the small subset of p53 target genes that play roles in ferroptosis sensitivity, including the cystine importer *SLC7A11*. As a result, increased levels of cysteine and glutathione (GSH) accumulate in cells from S47 humans and mice (*Jennis et al., 2016*; *Leu et al., 2019*). More recently, we showed that the ferroptotic defect in S47 mice leads to iron accumulation in their livers, spleens, and macrophages. We also showed that the S47 variant is positively associated with markers of iron overload in African Americans (*Singh et al., 2020*).

An emerging paradigm in the cancer literature is that tumor-predisposing genetic variants may paradoxically provide selection benefit to individuals, thus potentially explaining the frequency of these damaging alleles in the population. As an example, women carrying tumor-predisposing mutations in the *BRCA1* gene tend to be physically larger and show increased fertility (*Smith et al., 2012*). Here-in we show that mice carrying a knock-in S47 allele in a pure C57Bl/6 background show increased size, lean content (muscle), and metabolic efficiency, relative to littermate mice with WT p53. We report that mouse and human S47 cells show a significant increase in mTOR activity, due in part to increased mTOR-Rheb binding in S47 cells. We propose that these attributes may have led to a positive selection for this variant in sub-Saharan Africa. Our studies shed further light on the intricate regulation that exists between p53, mTOR activity, and metabolic output, in this case mediated by GSH and the control of cellular redox state.

## Results

### Higher basal mTOR activity in cells containing the S47 variant

We previously showed that human lymphoblastoid cells (LCLs) that are homozygous for the S47 variant of p53 are impaired for the transcriptional regulation of less than a dozen p53 target genes, compared to cells from individuals from the same family containing WT p53 (*Jennis et al., 2016*). We noted that several of these genes encode proteins that play roles in the negative regulation of mTOR (*Budanov and Karin, 2008*; *Feng et al., 2007*). We confirmed via qRT-PCR that S47 LCLs show modestly decreased expression of the p53 target genes *SESN1* and *PTEN*, and decreased transactivation of *PRKAB1*, relative to WT LCLs following cisplatin treatment (*Figure 1—figure supplement 1A and B*). These findings prompted us to assess basal mTOR activity in WT and S47 LCLs, and in MEFs from WT and S47 mice. To corroborate our findings, we also analyzed tissues from humanized p53 knock-in (Hupki) mice carrying WT and S47 alleles on a pure C57Bl/6 background, which we previously generated and characterized (*Jennis et al., 2016*). Western blot analysis of WT and S47 LCLs, along with multiple clones of WT and S47 MEFs, revealed increased p-S6K1 (Thr389) in S47 cells (*Figure 1A*). Following normalization to total S6K1, this increase ranged between two-

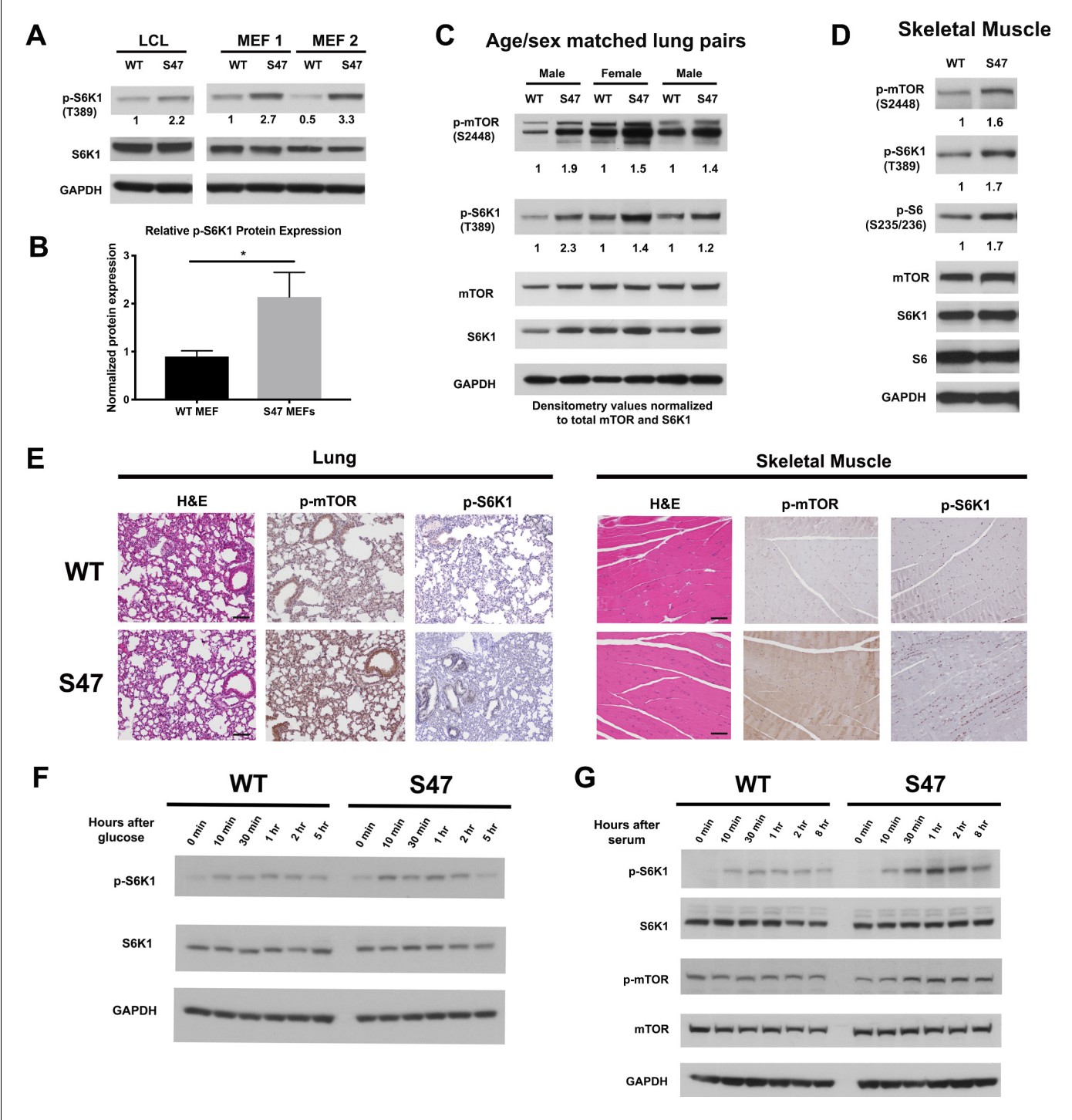

**Figure 1.** Increased markers of mTOR activity in S47 cells and tissues. (**A**) Western blot analyses reveal higher phospho-S6K1 expression in S47 LCLs and S47 MEFs; the latter were obtained from two separate embryos per genotype. (**B**) Densitometry quantification of phospho-S6K1 protein expression in WT and S47 MEFs from four independent experiments; all values normalized to total S6K1. Error bars represent standard error, (*) p value < 0.05. (**C**) Whole cell lysates were extracted from three WT and three S47 mouse lungs and analyzed by western blot for the proteins indicated. Pairs 1 and 3 are lungs isolated from male mice, pair 2 is lungs isolated from female mice. Densitometry quantification of phospho-S6K1 and phospho-mTOR was performed and normalized to total S6K1 and total mTOR protein expression, respectively. (**D**) Whole cell lysates were extracted from WT and S47 mouse skeletal muscle and analyzed as described above. Densitometry quantification of phospho-S6, phospho-S6K1, phospho-mTOR was performed and normalized to total S6, total S6K1 and total mTOR protein expression, respectively. (**E**) Immunohistochemical analysis of hematoxylin and eosin (H and E), phospho-mTOR and phospho-S6K1 in WT and S47 mouse lung and skeletal tissue. Data are representative of n = 4 mice per genotype. Scale

*Figure 1 continued on next page*

*Figure 1 continued*

bar represents 100 µm. (**F**) WT and S47 MEFs were grown in glucose-free media for 16 hr, then in media containing 4.5 g/L glucose. Samples were collected at indicated time points and analyzed by western blot for p-S6K1, total S6K1, and GAPDH. (**G**) WT and S47 MEFs were cultured in media containing 0.1% FBS for 16 hr, followed by media containing 10% serum and samples were collected at indicated time points. Cell lysates were extracted from samples and subjected to western blot analysis for the proteins indicated.
The online version of this article includes the following figure supplement(s) for figure 1:

**Figure supplement 1.** Altered metabolic markers in S47 cells and tissues.

and threefold (***Figure 1B***). We next compared mTOR activity in age- and sex-matched pairs of lung and muscle tissue from WT and S47 mice; we did this because mTOR activity is influenced by age and gender, and there is increased mTOR activity in female and older mice (***Baar et al., 2016***). We found increased levels of p-S6K1 (T389) and p-mTOR (Ser2448) in S47 lung and skeletal muscle, relative to WT tissues (***Figure 1C and D***). Immunohistochemical analysis of tissues from multiple age- and sex-matched WT and S47 mice confirmed these findings (***Figure 1E***). Interestingly, increased mTOR activity was not seen in all tissues of the S47 mouse (***Figure 1—figure supplement 1C***), and lung and skeletal muscle were the most consistently different between WT and S47. We also did not detect significant differences in p-AKT (Ser473) in WT and S47 lung tissue, suggesting that mTORC1 and not mTORC2 is likely responsible for the observed differences in mTOR activity (***Figure 1—figure supplement 1D***).

We next sought to test the kinetics of mTOR activation in WT and S47 cells by subjecting early passage WT and S47 MEFs to nutrient deprivation, followed by monitoring of mTOR activation markers after nutrient restoration using antisera to p-S6K1 and p-mTOR. Glucose deprivation experiments revealed consistent albeit modest increases in p-S6K1 following glucose refeed in S47 MEFs, compared to WT (***Figure 1F***). Serum deprivation experiments revealed more pronounced results. For serum deprivation, we subjected three independent cultures each of WT and S47 MEFs to 0.1% serum for 16 hr, followed by 10% serum, after which total and phospho -S6K1 and -mTOR were monitored in a time course. S47 cells consistently showed increased induction of markers of mTOR activation after serum re-feed compared to WT cells (***Figure 1G***). We next performed amino acid deprivation experiments; these likewise showed increased response in S47 cells (***Figure 1—figure supplement 1E***). Combined densitometry results from all forms of nutrient deprivation revealed an approximately two- to threefold increase in p-S6K1 in S47 cells following nutrient restoration at 30 or 60 min (***Figure 1—figure supplement 1F***).

Given that mTOR plays a role in autophagy inhibition (***Jung et al., 2010***; ***White et al., 2011***), we wondered whether basal autophagy or autophagic flux might be decreased in S47 cells. We were unable to see any differences in the steady state levels of LC3B or the autophagy adaptor protein p62$^{SQSTM1}$ in WT and S47 MEFs or tissues, either at steady state (***Figure 1—figure supplement 1G***) or following HBSS treatment to induce autophagy (***Figure 1—figure supplement 1H***). Likewise, we failed to see differences in autophagic flux (conversion of LC3-I to LC3-II when the lysosome is inhibited (***Figure 1—figure supplement 1I***) between WT and S47 cells, or in cell viability after HBSS treatment (***Figure 1—figure supplement 1J***). Therefore, while markers of mTOR activity are clearly increased in S47 cells and tissues, this does not appear to be accompanied by alterations of basal or induced autophagy.

## Enhanced mitochondrial function and glycolysis in S47 cells

To determine the functional consequences of the increased markers of mTOR activity in S47 cells, we used a Seahorse BioAnalyzer to assess the oxygen consumption rate (OCR), as well as basal and compensatory glycolytic rate in WT and S47 MEFs and LCLs. Seahorse analyses revealed that S47 LCLs show increased OCR under stressed conditions compared to WT (***Figure 2A***). These analyses also revealed that human S47 LCLs and mouse S47 MEFs show increased basal and compensatory glycolysis, compared to WT cells (***Figure 2B and C***). We next assessed glucose and glutamine consumption using a Yellow Springs Instrument (YSI) Analyzer. These analyses revealed that S47 LCLs and MEFs show significantly increased consumption of glucose and glutamine, along with increased production of lactate and glutamate, compared to WT cells (***Figure 2D and E***); again, multiple independent MEF lines were analyzed. Interestingly, LCLs from individuals heterozygous for the S47

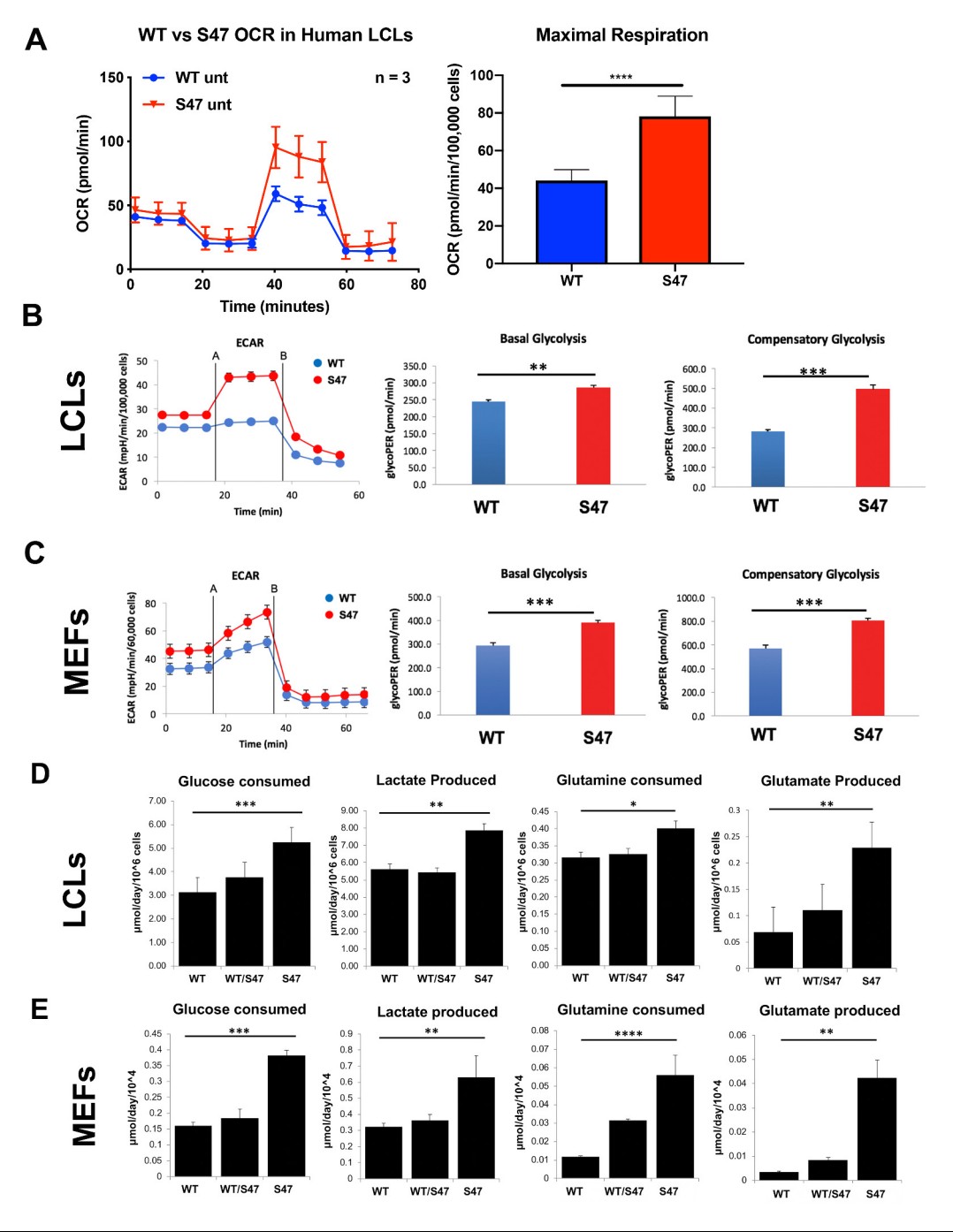

**Figure 2.** Increased metabolism in S47 cells compared to WT cells. (**A**) Oxygen consumption rates (OCR) in WT and S47 LCLs were assessed using the Seahorse XF Mito Stress Test. OCR was measured first in basal conditions, and following injection of oligomycin, FCCP and rotenone/antimycin. The bar graph depicts maximal OCR after FCCP injection at ~40 min timepoint; data are representative of three independent experiments performed with at least six technical replicates, presented as mean ± SD. (**B–C**) Basal and compensatory glycolysis in WT and S47 LCLs (**B**) and MEFs (**C**) were assessed using the Seahorse Glycolytic Rate Assay. Basal glycolysis is first measured, followed by treatment of cells with rotenone/antimycin and 2-deoxy-D-glucose (2-DG). The bar graph depicts basal glycolysis at ~1 min timepoint and compensatory glycolysis after antimycin/rotenone injection at ~22 min timepoint; data are representative of three independent experiments performed with at least 10 technical replicates. Bar graphs are presented as mean ± SD. (**D–E**) Consumption of glucose and glutamine from media and production of lactate and glutamate were analyzed from homozygous WT, heterozygous WT/S47 and

*Figure 2 continued on next page*

*Figure 2 continued*

homozygous S47 human LCLs (**D**) and primary MEFs (**E**) using a YSI-7100 Bioanalyzer. Means and SEM are shown (n = 5).

The online version of this article includes the following source data and figure supplement(s) for figure 2:

**Figure supplement 1.** Increased metabolism in S47 MEFs but no differences in mitochondrial content in WT and S47 cells.

**Figure supplement 1—source data 1.** Metabolomics source data.

variant (S47/WT), and MEFs from S47/WT mice, showed values typically intermediate between homozygous WT and S47 cells (*Figure 2D and E*). We next performed metabolic flux analyses in WT and S47 cells using $^{13}$C-labeled glucose. Analysis of $^{13}$C$_6$-glucose tracing in WT and S47 MEFs provided evidence for a higher contribution of glucose carbon into the TCA cycle in S47 cells compared to WT cells, as evidenced by increased labeling of citrate, malate, aspartate, and glutamate in S47 MEFs (*Figure 2—figure supplement 1A–D*). We reasoned that one possibility for the increased metabolism in S47 cells might be due to increased mitochondrial content, which is regulated by mTOR (*Morita et al., 2013*). However, MitoTracker analyses and western blotting for mitochondrial proteins revealed no obvious increase in mitochondrial content in S47 cells (*Figure 2—figure supplement 1E and F*). We find no evidence that S47 LCLs and MEFs proliferate more quickly than WT cells (*Jennis et al., 2016*); this raises the possibility that this increased nutrient consumption may be used for biomass instead of proliferation.

Because mTOR is known to regulate mitochondrial function (*Morita et al., 2013*; *Schieke et al., 2006*; *Ye et al., 2012*), we next assessed the impact of mTOR inhibitors on mitochondrial function in WT and S47 cells. Seahorse analysis of WT and S47 LCLs revealed that S47 cells are less susceptible to inhibition of oxygen consumption rate and maximal respiration by the mTOR inhibitors rapamycin (*Figure 3A and B*) and Torin1 (*Figure 3C and D*). This finding was not due to altered efficacy of each inhibitor, as evidenced by similar decreases in p-mTOR and p-S6 in WT and S47 cells following treatment with rapamycin (*Figure 3—figure supplement 1A*) and Torin1 (*Figure 3E*), and by the finding that very high concentrations of Torin1 were able to inhibit oxygen consumption equally well in both WT and S47 cells (*Figure 3—figure supplement 1B*). These data support the possibility of an S47 dependent, but mTOR-independent, effect on OCR as well.

## Increased mTOR activity in S47 is due to increased mTOR-Rheb interaction

We next sought to identify the mechanism underlying increased mTOR activity in S47 cells. Unfortunately, although we identified decreased mRNA levels of some mTOR regulators in S47 cells, we found no evidence for significant differences at the protein level of any p53-regulated mTOR regulators in steady state MEFs (*Figure 4—figure supplement 1A*) or following treatment with Nutlin to induce p53 (*Figure 4—figure supplement 1B*). Therefore, we turned to a key regulator of mTOR activity, the small GTPase Rheb, which binds and activates mTOR (*Long et al., 2005*). We monitored the mTOR-Rheb association in WT and S47 MEFs using the technique of proximity ligation assay (PLA), which quantitatively detects protein-protein interactions. PLA experiments revealed that there were consistently increased mTOR-Rheb complexes in S47 cells, compared to WT; this was true in multiple replicates, in multiple MEF clones, and using single antibody controls that showed no signal (*Figure 4A*). Quantification of multiple experiments revealed an approximately two-fold increase in mTOR-Rheb complexes in S47 cells compared to WT (*Figure 4B*), which is consistent with all of our analyses of mTOR activity.

One regulator of the mTOR-Rheb interaction is the cytosolic enzyme GAPDH. This enzyme binds to Rheb and sequesters it from mTOR in cultured cells, in a manner that is regulated by glucose levels (*Lee et al., 2009*). First, we confirmed that the interaction between Rheb and GAPDH is detectable in the skeletal muscle of mice using IP-western, and moreover that this interaction is regulated by glucose (*Figure 4—figure supplement 1C*). Next, we performed immunoprecipitation (IP)-western of Rheb in skeletal muscle extracts from WT and S47 mice; we found that there was increased mTOR, and significantly decreased GAPDH, in Rheb IPs from S47 skeletal muscle compared to WT (*Figure 4C*). The combined data from three independent IPs of Rheb in WT and S47 skeletal muscle

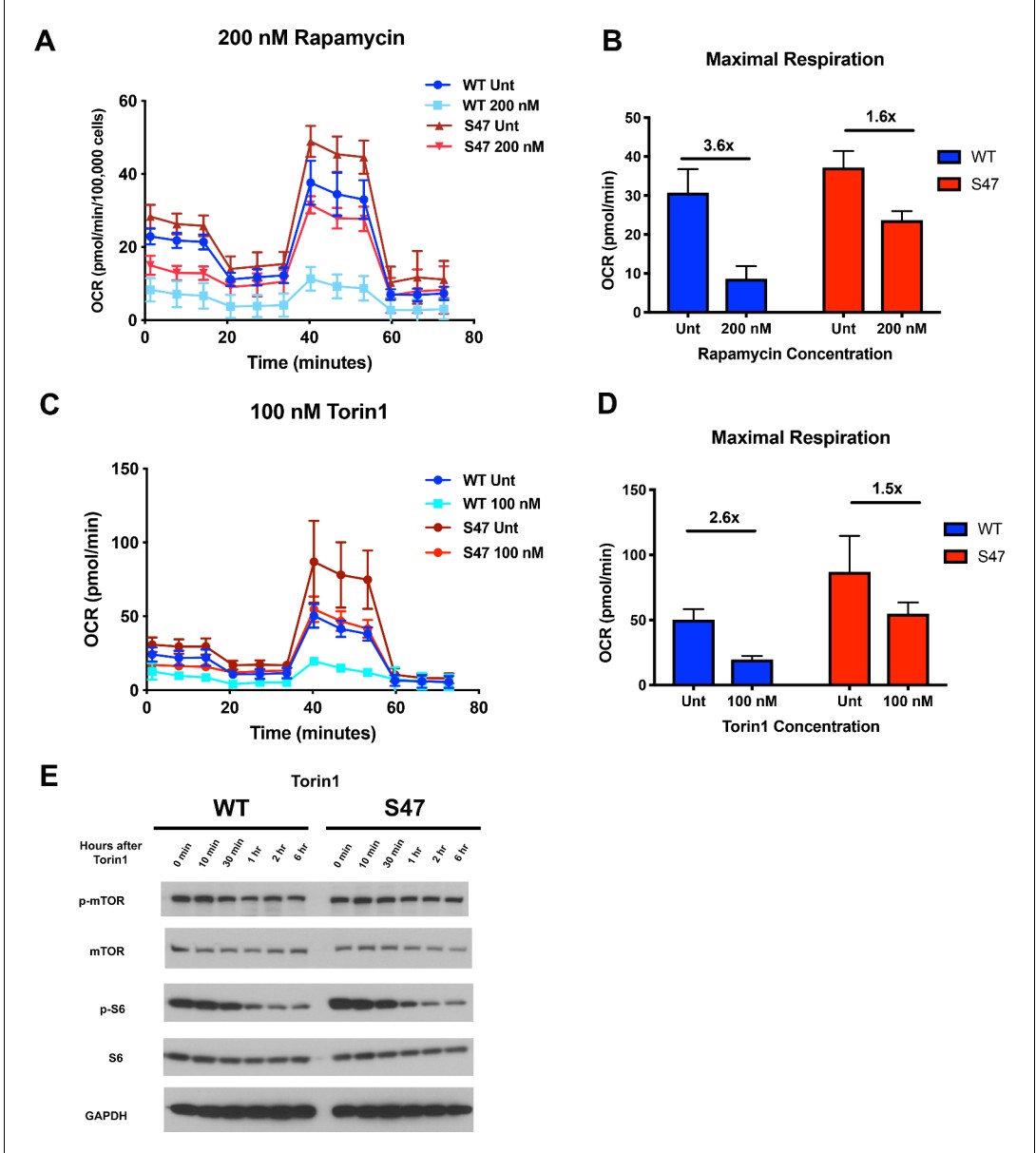

**Figure 3.** S47 mitochondria show decreased sensitivity to mTOR inhibition. (**A**) Oxygen consumption rate (OCR) as measured by the Seahorse XF Mito Stress Test in WT and S47 LCLs treated with 200 nM of rapamycin for 24 hr. (**B**) Bar graph depicts maximal OCR after FCCP injection at ~40 min timepoint; fold changes between rapamycin treated and untreated samples are shown. Data are representative of two independent experiments performed with at least 10 technical replicates. (**C**) OCR as measured by the Seahorse XF Mito Stress Test in WT and S47 LCLs treated with 100 nM of Torin1 for 24 hr. (**D**) Bar graph depicts maximal OCR after FCCP injection at ~40 min timepoint; fold changes between Torin1 treated and untreated samples are shown. Data are representative of two independent experiments performed with at least eight technical replicates. (**E**) WT and S47 LCLs were treated with 100 nM of Torin1, harvested at indicated time points after treatment and analyzed for the shown mTOR markers via western blot. The online version of this article includes the following figure supplement(s) for figure 3:

**Figure supplement 1.** Attenuated response to mTOR inhibition in S47 cells.

revealed an approximately twofold decrease in the amount of GAPDH co-precipitating with Rheb in S47 skeletal muscle compared to WT (p<0.05, *Figure 4D*). In contrast, there were no differences in the levels of mTOR, Rheb and GAPDH in these extracts (*Figure 4C* – WCL, *Figure 4—figure supplement 1D*). Consistent with our IP-western findings, PLA analyses corroborated that the GAPDH-Rheb association is markedly decreased in S47 MEFs relative to WT MEFs (*Figure 4A*). Confocal microscopy analyses revealed no significant differences in the cellular localization of Rheb at the

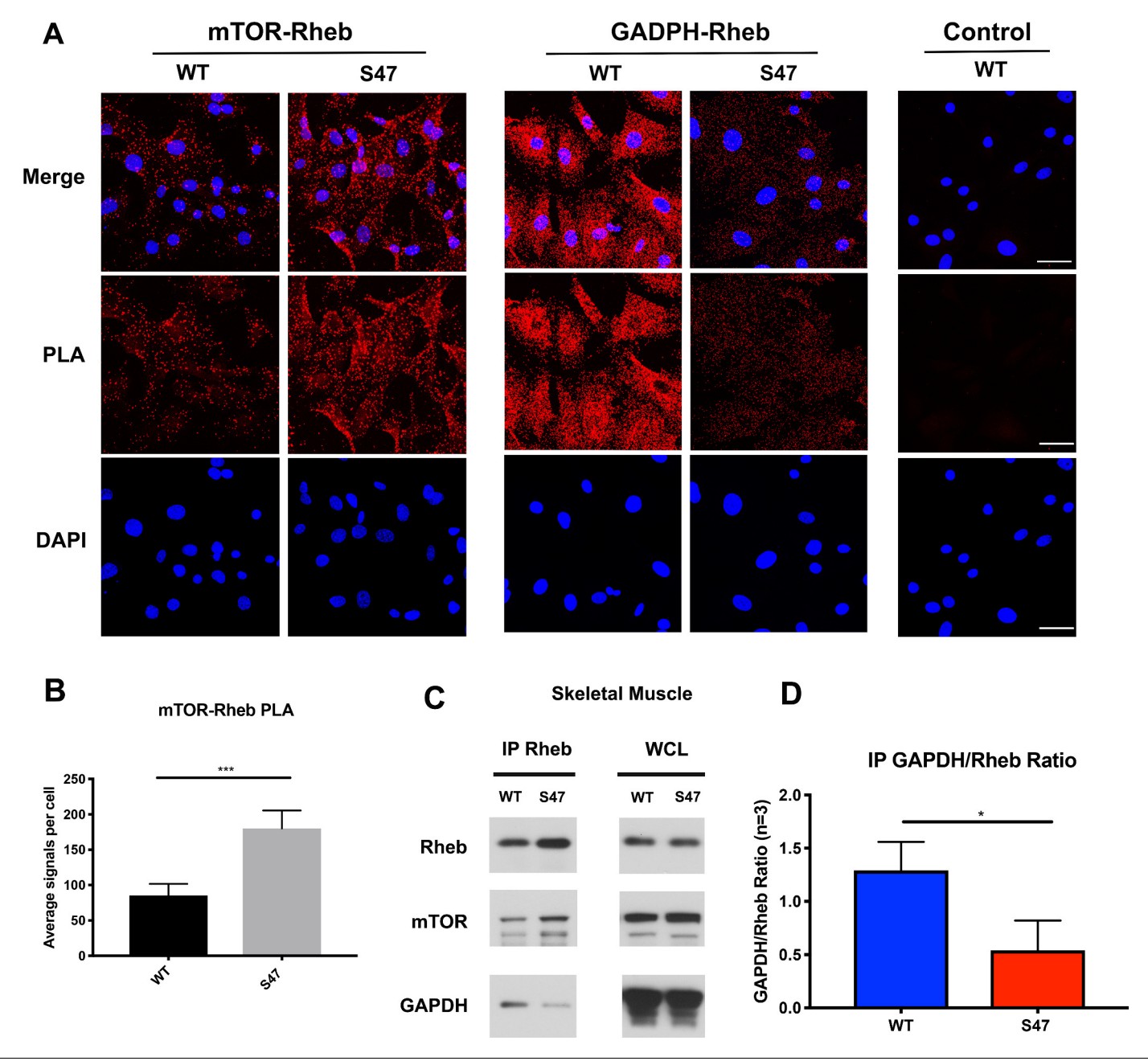

**Figure 4.** Increased mTOR-Rheb binding in S47 cells is due to decreased GAPDH-Rheb binding. (**A–B**) An in situ proximity ligation assay (PLA) was performed in WT and S47 MEFs. Each red dot represents an interaction between mTOR-Rheb or GAPDH-Rheb as indicated; scale bar represents 50 µm. The samples were counterstained with DAPI to detect nuclei. Cells stained in the absence of one primary antibody were used as a negative control. (**B**) Quantification of the mTOR-Rheb interactions, measured as the average number of PLA signals per nuclei. Data were quantified by counting the number of cells in five random fields per experimental condition. (\*\*\*) p-value<0.001, Student's t-test. (**C**) Lysates extracted from WT and S47 skeletal tissue were immunoprecipitated with anti-Rheb. The amount of co-precipitating mTOR and GAPDH, as well as immunoprecipitated Rheb, were assessed by western blot. Whole cell lysate (WCL) is shown on the right. (**D**) Quantification of the amount of GAPDH bound to Rheb, divided by total Rheb pulled down, in WT and S47 skeletal tissue, n = 3 independent experiments, (\*) p-value<0.05.

The online version of this article includes the following figure supplement(s) for figure 4:

**Figure supplement 1.** No differences in the level of mTOR regulators in WT and S47 cells.

lysosome, as assessed by LAMP1 localization, nor were there any differences in TSC2 localization at the lysosome in WT and S47 MEFs (*Figure 4—figure supplement 1E*). The combined data support the premise that the increased mTOR activity in S47 cells may be due to increased Rheb-mTOR association, caused by a decreased association of Rheb with GAPDH. We sought to test this hypothesis, and identify the underlying mechanism.

GAPDH is a multi-functional enzyme that is known to be sensitive to redox status (*Brandes et al., 2009*; *Chernorizov et al., 2010*). We hypothesized that the increased glutathione (GSH) levels in S47 cells (*Leu et al., 2019*) might alter the redox state of GAPDH and impact its ability to bind to Rheb. We first verified that lung tissue and skeletal muscle from S47 mice possess increased GSH compared to WT tissues, as assessed by an increased ratio of reduced versus oxidized glutathione (GSH:GSSG) (*Figure 5A*). We then validated that the GSH alkylating agent diethylmaleate (DEM) could successfully decrease the level of GSH, and the GSH/GSSG ratio, in these cells (*Figure 5A*); these findings are consistent with previously published findings by our group (*Leu et al., 2019*). Notably, DEM treatment of immortalized S47 MEFs (iMEFs) caused a dramatic decrease in markers of mTOR activity (p-mTOR and p-S6K1), showing that modulation of GSH, even for as little as five hours, can impact mTOR activity (*Figure 5B*). Moreover, we found that the impact of DEM on mTOR activity was enhanced by the addition of glutamate, which decreases cystine import through system Xc(-), leading to decreased level of GSH (*Figure 5B*).

We next sought to test the hypothesis that the redox state of GAPDH was altered in WT and S47 cells. Toward this end, we employed cross-linking experiments using the cysteinyl cross-linking agent bismaleimidohexane (BMH), which cross-links cysteine residues within 13 Å by covalently conjugating free (reduced) sulfhydryl groups (*Green et al., 2001*). We treated freshly isolated lung and skeletal muscle lysates from WT and S47 mice, and from immortalized WT and S47 MEFs (iMEFs), with BMH. Cysteinyl-crosslinked proteins were resolved on SDS-PAGE gels and compared to untreated extracts. Notably, we found consistent differences in GAPDH cross-linking patterns in S47 samples compared to WT, as evidenced by altered mobility of GAPDH on SDS-PAGE of BMH-treated samples (*Figure 5C*). Moreover, the altered mobility of GAPDH in S47 cells could be reversed following glutathione depletion by DEM treatment (*Figure 5C*) or by the compound BSO (buthionine sulfoximine; *Figure 5—figure supplement 1A*) which inhibits GSH biosynthesis. Treatment with the compound erastin, which inhibits the system Xc(-) cystine transporter and decreases GSH, also led to altered mobility of GAPDH in S47 and WT cells (*Figure 5—figure supplement 1B*). We next tested the impact of modulating GSH on the interaction between Rheb and GAPDH in WT and S47 cells using both IP-western and PLA. By IP-western we found that supplementation of culture media with exogenous GSH decreased the Rheb-GAPDH interaction in WT cells; conversely, depleting free GSH using either BSO or DEM increased the Rheb-GAPDH interaction in S47 cells (*Figure 5D*). These findings were corroborated using PLA, which revealed that depleting GSH using either DEM or BSO completely restores GAPDH-Rheb complex formation and mTOR-Rheb complex formation in S47 cells, to levels equivalent to WT cells (*Figure 5E and F*; *Figure 5—figure supplement 1C*). The combined data support the conclusion that the increased GSH pool in S47 cells affects the status of reactive cysteines in GAPDH, and the ability of this protein to bind and sequester Rheb, thereby leading to increased Rheb-mTOR interaction and increased mTOR activity in S47 cells.

## Enhanced metabolic efficiency of S47 mice

mTOR is known to regulate body mass and muscle regeneration (*Laplante and Sabatini, 2012*; *Yoon, 2017*). We therefore next assessed body weight and fat/lean content in age-matched male mice of WT and S47 genotypes. We also tracked body weight with age of multiple male and female sibling littermate mice of WT/WT, WT/S47, and S47/S47 genotypes in our colony. S47 mice showed significantly increased weight with time, compared to WT/WT and WT/S47 sibling littermates (*Figure 6—figure supplement 1A*). Body composition analysis using nuclear magnetic resonance revealed that S47 mice had significantly increased fat and lean content, compared to WT mice (*Figure 6A*; *Figure 6—figure supplement 1B*). We next analyzed the metabolic activities of WT and S47 mice using a comprehensive lab animal monitoring system (CLAMS) over the course of 48 hr. In this analysis, S47 mice showed comparable locomotor activity to WT mice but reduced food intake, oxygen consumption, and heat production (*Figure 6B*). These CLAMS data suggested that S47 mice might possess enhanced metabolic efficiency compared to WT mice, and prompted us to assess the response of WT and S47 mice to exercise challenge.

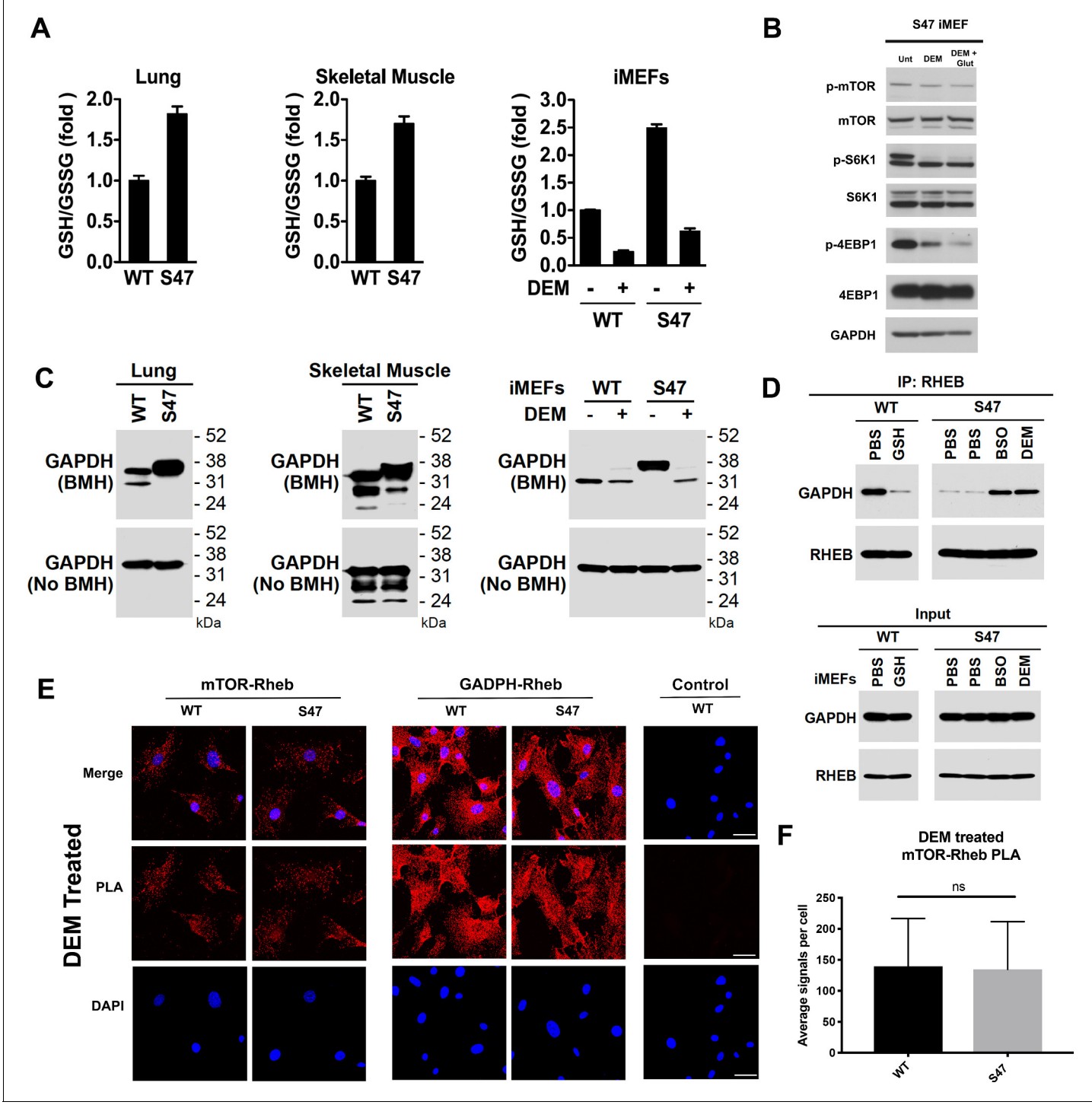

**Figure 5.** Increased glutathione drives decreased GAPDH-Rheb binding in S47 cells. (**A**) WT and S47 lung (left) and skeletal muscle (center) were assessed for GSH/GSSG ratio (mean ± SD, *n* = 3). WT and S47 immortalized MEFs (iMEFs), either untreated or treated with 50 µM DEM for 5 hr, were analyzed for GSH/GSSG ratio (mean ± SD, n = 4). (**B**) WT and S47 iMEFs were untreated or treated with 50 µM of DEM or 50 µM of DEM + 0.5 mM glutamate for 5 hr and protein lysates were analyzed by western blot for indicated mTOR markers. (**C**) Whole cell lysates were extracted from WT and S47 mouse lung (left) and skeletal (center) tissue. Proteins were cross-linked with BMH, resolved by SDS/PAGE, and detected by western blotting with a GAPDH specific antibody (Top). Untreated protein lysates were analyzed by western blot analysis for total GAPDH (Bottom). WT and S47 iMEFs were treated with 50 µM of DEM for 5 hr and protein lysates were analyzed as described (right). (**D**) WT cells were treated with PBS or 3 mM GSH for 24 hr. S47 cells were treated with PBS for 24 hr, PBS for 5 hr, 100 µM BSO for 24 hr, or 50 µM DEM for 5 hr. IP of the lysates with anti-Rheb followed by western analysis for associated GAPDH and Rheb (top panel). The same lysates were analyzed by western blotting for GAPDH and Rheb (bottom

*Figure 5 continued on next page*

*Figure 5 continued*

panel). (**E–F**) Proximity ligation analysis (PLA) was performed in WT and S47 MEFs treated with 50 µM of DEM for 5 hr and analyzed as described in *Figure 4A–B*. Scale bar is 30 µm.

The online version of this article includes the following figure supplement(s) for figure 5:

**Figure supplement 1.** Glutathione depletion by BSO alters GAPDH cross-linking and the GAPDH-Rheb interaction.

We subjected WT and S47 mice to treadmill exercise with increasing intensity over time. For this analysis, we studied eight age-matched male mice of each genotype during a 50 min forced exercise at increasing speed and slope. During this time course, oxygen consumption and serum metabolites were quantified. Consistent with our CLAMs experiment, S47 mice started with lower basal $VO_2$ and exhibited generally lower $VO_2$ for the work being performed; however, as they approached the final, most strenuous point of the exercise, the $VO_2$ values in WT and S47 converged, so the $VO_2$ range for S47 mice was significantly greater than WT mice (*Figure 6C–E*). Analysis of serum metabolites and proteins before and after exercise revealed decreased lactate dehydrogenase (LDH) levels in the sera of S47 mice, which is indicative of decreased muscle damage in S47 mice compared to WT (*Figure 6F*). Additionally, we found that Ki-67 staining in S47 skeletal muscle was consistently increased relative to WT muscle, suggesting an enhanced ability for S47 muscle to recover (*Figure 6—figure supplement 1C*). In these tissues, we found no differences in p53 level or markers of mitochondrial content (*Figure 6—figure supplement 1D and E*), nor were there other differences in other serum metabolites between WT and S47 mice (*Figure 6—figure supplement 1F*). These exercise data, like the CLAMS data, point to increased metabolic efficiency in S47 mice relative to WT mice. To address this further, we analyzed a small cohort of WT and S47 mice on a continuous strenuous treadmill run. Although the numbers are small, we found that three out of four WT mice failed to complete a 60 min strenuous run, while three out of four S47 successfully completed this run (*Figure 6—figure supplement 1G*).

## Discussion

In this study, we report that cells and mice with the S47 variant of p53 have increased mTOR activity and evidence for increased metabolic efficiency. The animals also display increased mass and signs of superior fitness. Our data support the premise that the enhanced mTOR activity is due, at least in part, to the higher levels of GSH in S47 cells and tissues. The increased GSH results in impaired ability of the redox sensitive protein GAPDH to bind to Rheb. This leads to greater mTOR-Rheb binding, resulting in increased mTOR activity in S47 cells and tissues. These data indicate that, along with pH (*Walton et al., 2018*), cellular redox status can also regulate mTOR activity, in a manner controlled by p53. We show that oxidative metabolism in S47 cells is less sensitive to mTOR inhibitors, thus tying these two phenotypes together; this is not surprising, as a link between mTOR and a number of cellular metabolic processes is well known (*Morita et al., 2013*; *Schieke et al., 2006*).

We see evidence for increased mTOR activity only in certain tissues of the S47 mouse, so the metabolic impact of this genetic variant appears to be influenced by tissue type and cellular environment. At present, we do not know if this tissue specificity is due to differences in GSH level, or to altered mTOR-Rheb or GAPDH-Rheb interactions in different tissues, or to other parameters. We also see evidence for some unexpected findings regarding the increase in mTOR activity in S47 cells: given that mTOR negatively regulates autophagy (*Jung et al., 2010*), we expected to see differences in steady state autophagy or autophagic flux in WT and S47 cells, but we found no evidence for this. This finding may be due to the rather complex relationship between mTOR and autophagy (*Jung et al., 2010*; *White et al., 2011*), and/or that other signaling pathways regulate autophagy aside from mTOR, including the PI3K pathway, GTPases, and calcium (*Yang et al., 2005*).

The increased lean content in S47 mice likely contributes to the increased fitness observed in these mice. Human studies have shown that mTOR activation is crucial for human muscle protein synthesis (*Dickinson and Rasmussen, 2011*). Treatment with the well-studied mTOR inhibitor rapamycin blocks the effects of amino acid ingestion on mTOR activity and leads to decreased protein synthesis in human skeletal muscle (*Dickinson and Rasmussen, 2011*; *Drummond et al., 2009*). Additionally, mTOR signaling driven through IGF-1 plays a key role in promoting muscle hypertrophy

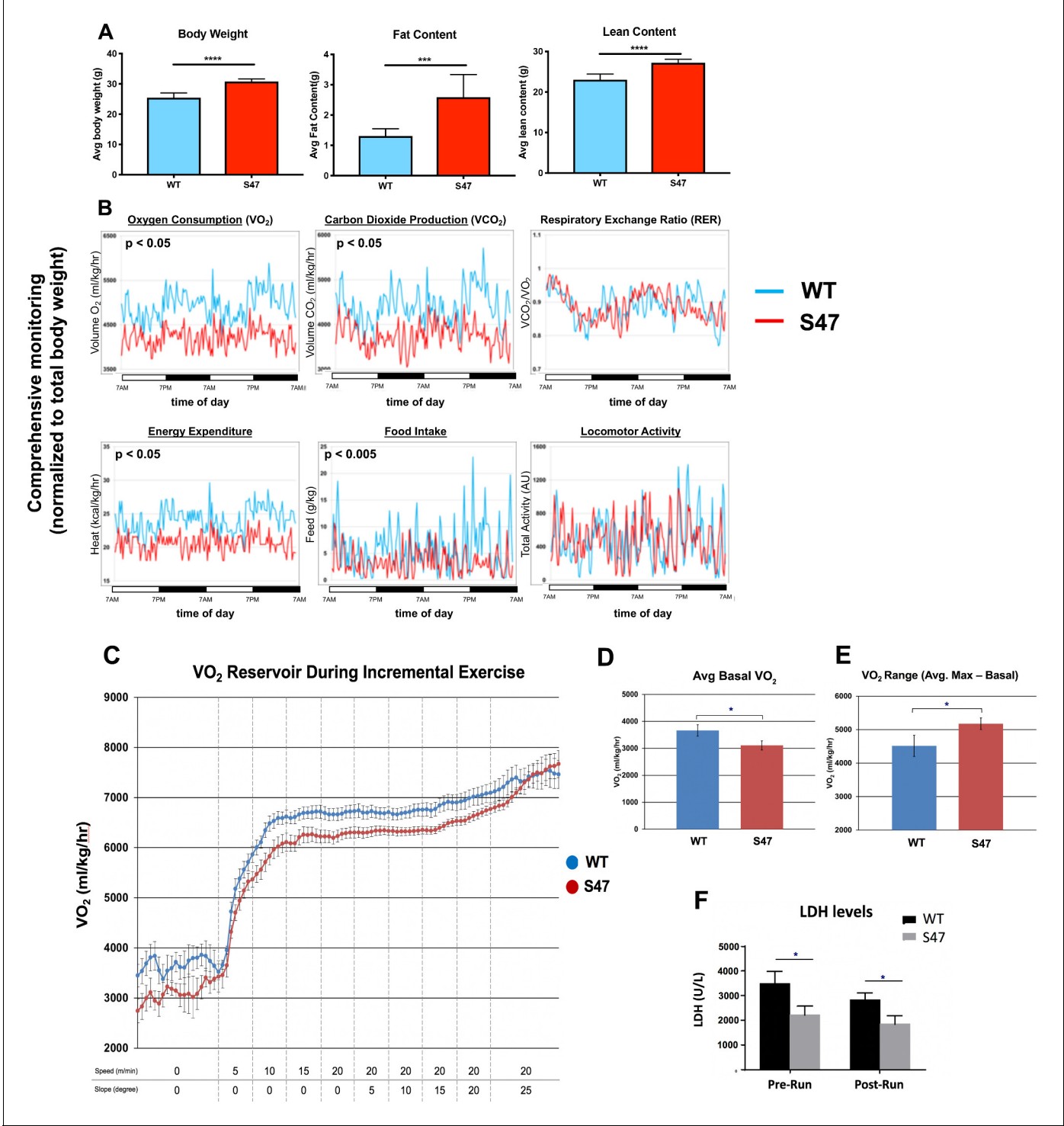

**Figure 6.** Increased size and improved metabolic efficiency in S47 mice. (**A**) Nuclear magnetic resonance (NMR) studies revealed S47 mice have increased body weight, increased fat content and increased lean content, n = 7 WT mice, n = 8 S47 mice. (***) p-value<0.001, (****) p-value<0.0001. Bar graphs are presented as mean ± SD. (**B**) Changes in metabolic parameters for WT mice (blue) and S47 mice (red) were determined by using the Comprehensive Lab Animal Monitoring System for 48 hr. Parameters assessed includes oxygen consumption, carbon dioxide production, respiratory exchange rate, energy expenditure, total food intake, and locomotor activity. The data are representative of five 6-week old male mice per genotype and are normalized to total body weight. (**C**) WT and S47 mice (n = 6–7) were subjected to a treadmill study of increasing intensity over time. Oxygen consumption ($VO_2$) is normalized to body mass. (**D**) Mean basal $VO_2$ in WT and S47 mice. (**E**) $VO_2$ range in WT and S47 mice determined by subtracting the mean basal $VO_2$ from the $VO_2$ max, obtained during the most strenuous point of exercise at the tail end of the treadmill study. (**F**) Lactate

*Figure 6 continued on next page*

*Figure 6 continued*

dehydrogenase (LDH) levels measured in the serum of WT and S47 mice obtained before and after the treadmill study. (*) p-value<0.05, Student's t-test.

The online version of this article includes the following source data and figure supplement(s) for figure 6:

**Source data 1.** Metabolic efficiency source data 1.
**Source data 2.** Metabolic efficiency source data 2.
**Source data 3.** Metabolic efficiency source data 3.
**Figure supplement 1.** Serum metabolites and protein markers pre- and post- exercise.

(*Coleman et al., 1995*; *Musarò et al., 2001*; *Vandenburgh et al., 1991*). One caveat of this study, however, is that we do not directly demonstrate that the increased mTOR activity in S47 mice is causing their increased lean content or superior performance on treadmill assays. Transient treatment with mTOR inhibitors elicits highly complex and often contrasting effects on energy expenditure and treadmill performance, likely due to the existence of feedback loops and the effect of inhibitors on multiple organ systems in the mouse. As just two examples: rapamycin has shown contrasting effects on energy expenditure in animals, depending on how long mice are treated (*Fang et al., 2013*); similarly, treatment of mice with rapamycin has shown limited impact on treadmill endurance, despite causing decreased expression of genes involved in mitochondrial biogenesis and oxidative phosphorylation in the muscle (*Ye et al., 2013*). Possibly, the most consistent findings in the literature reflect the general consensus that mTORC1 is involved in mechanisms that drive increased muscle mass (*Goodman, 2019*) and that heightened mTOR activity leads to enhanced muscle recovery after exercise (*Song et al., 2017*; *Yoon, 2017*). It remains to be tested if these are the pathways affected in S47 mice.

We hypothesize that the more efficient metabolism and enhanced fitness provided by the S47 variant may have once provided carriers with a bio-energetic advantage in Sub-Saharan western Africa, where this variant is most common. For example, those carrying the S47 SNP may have possessed superior athletic prowess and/or ability to withstand famine (see model, *Figure 7*). This metabolic advantage may explain the high frequency of this genetic variant in sub-Saharan Africa, despite the fact that it predisposes individuals to cancer later in life. Another positive selection for this variant in Africa may include an improved ability to withstand malaria infection: we recently reported that the S47 variant alters the immune micro-environment in mice and confers improved response to the malaria toxin hemozoin (*Singh et al., 2020*).

Our findings provide further support for the growing premise that some tumor suppressor genetic variants may provide evolutionary selection benefit (*Vicens and Posada, 2018*). For example, women who carry the BRCA1/2 mutation exhibit increased size and enhanced fertility when compared to controls (*Smith et al., 2012*). Similarly, people with Li Fraumeni syndrome who inherit germline mutations in *TP53*, as well as mice with tumor-derived germline mutations in *Tp53*, demonstrate increased fitness endurance (*Wang et al., 2013*); however, this is due to increased mitochondrial content, which we do not see in S47 cells. A common genetic variant in *TP53* at codon 72, encoding proline at amino acid 72, confers increased longevity while conversely causing increased cancer risk (*Zhao et al., 2018*). In contrast, the arginine 72 variant of p53 induces increased expression of LIF, which improves fecundity (*Kang et al., 2009*). The take home message from all these studies is that the diverse roles of tumor suppressor proteins like p53 in metabolism, fertility, and fitness may allow for positive selection for certain variants, even at the expense of increased cancer risk. In mice, this increased cancer risk occurs quite late in life, well past reproductive selection (12–18 months). More needs to be done to analyze cancer risk in S47 humans. A more comprehensive understanding of the function of tumor suppressor genetic variants, including the S47 SNP, will enable improved understanding of cancer risk, along with superior personalized medicine approaches, with the ultimate goal of improving clinical outcomes and survival of people who carry this variant.

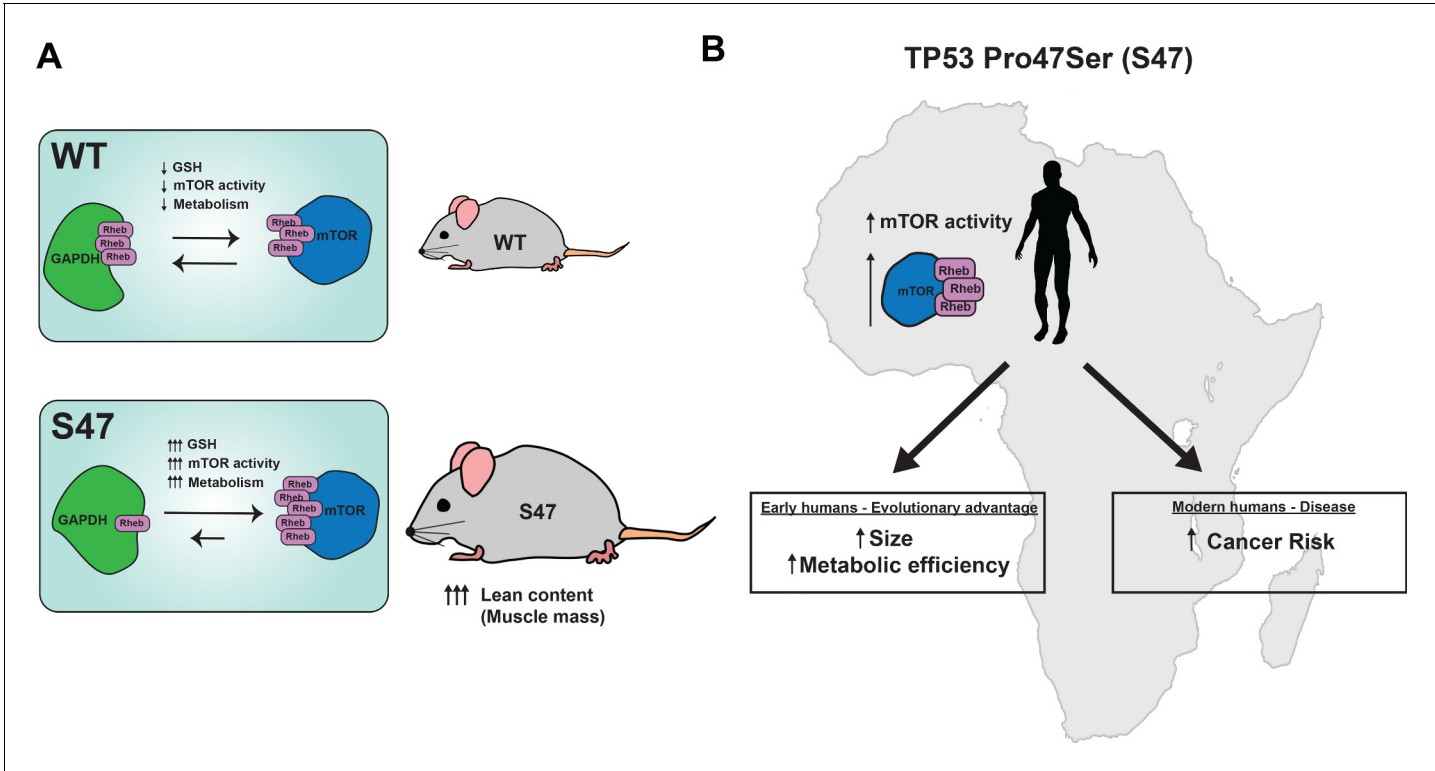

**Figure 7.** Proposed model. (**A**) Proposed model of how S47 contributes to increased metabolism. The elevated levels of GSH alter the redox state of the S47 cell, in turn affecting GAPDH conformation and impairing GAPDH-Rheb binding. This results in increased mTOR-Rheb binding, leading to increased mTOR activity and resulting an overall increase in metabolism in S47 mice, as seen by increased fat and lean content. (**B**) Broad impact of S47 variant: although providing an adaptive advantage to individuals residing in sub-Saharan Africa at one point in time, it now predisposes modern humans with this SNP to cancer.

# Materials and methods

## Key resources table

| Reagent type (species) or resource | Designation | Source or reference | Identifiers | Additional information |
|---|---|---|---|---|
| Strain, strain background (*M. musculus*) | C57Bl/6; P47/S47 Hupki | *Jennis et al., 2016* | | PMID:27034505 |
| Cell lines (*M. musculus*) | WT/S47 Primary MEFs | *Jennis et al., 2016* | | PMID:27034505 |
| Cell lines (*Homo-sapiens*) | WT/S47 LCLs | Coriell Institute | GM18870 GM18871 GM18872 | PMID:27034505 |
| Antibody | Anti-phospho-mTOR (Ser2448) (Rabbit polyclonal) | Cell Signaling Technologies | Cat# 2971 RRID:AB_330970 | WB (1:1000) IHC (1:100) |
| Antibody | Anti-mTOR (7C10) (Rabbit monoclonal) | Cell Signaling Technologies | Cat# 2983 RRID:AB_2105622 | WB (1:1000) PLA (1:500) |

*Continued on next page*

*Continued*

| Reagent type (species) or resource | Designation | Source or reference | Identifiers | Additional information |
|---|---|---|---|---|
| Antibody | Anti-phospho-S6K1 (Thr389) (Rabbit polyclonal) | Cell Signaling Technologies | Cat# 9205 RRID:AB_330944 | WB (1:1000) |
| Antibody | Anti-phospho-S6K1 (Thr421) (Rabbit polyclonal) | Thermo Fisher Scientific | Cat# PA5-37733 RRID:AB_2554341 | IHC (1:100) |
| Antibody | Anti-S6K1 (Rabbit polyclonal) | Cell Signaling Technologies | Cat# 9202 RRID:AB_331676 | WB (1:1000) |
| Antibody | Anti-GAPDH (14C10) (Rabbit monoclonal) | Cell Signaling Technologies | Cat# 2118 RRID:AB_561053 | WB (1:10,000) PLA (1:1000) |
| Antibody | Anti-TFAM (Rabbit polyclonal) | Abcam | Cat# ab131607 RRID:AB_11154693 | WB (1:2000) |
| Antibody | Anti-MTCO1 (Mouse monoclonal) | Abcam | Cat# ab14705 RRID:AB_2084810 | WB (1:2000) |
| Antibody | Anti-SDHA (Rabbit polyclonal) | Cell Signaling Technologies | Cat# 5839 RRID:AB_10707493 | WB (1:1000) |
| Antibody | Anti-Tom20 (F10) (Mouse monoclonal) | Santa Cruz Biotechnology | Cat# sc-17764 RRID:AB_628381 | WB (1:100) |
| Antibody | Anti-phospho-Akt (Ser473) (Rabbit monoclonal) | Cell Signaling Technologies | Cat# 4060 RRID:AB_2315049 | WB (1:1000) |
| Antibody | Anti-p62 (Rabbit polyclonal) | Cell Signaling Technologies | Cat# 5114 RRID:AB_10624872 | WB (1:1000) |
| Antibody | Anti-LC3B (D11) (Rabbit monoclonal) | Cell Signaling Technologies | Cat# 3868 RRID:AB_2137707 | WB (1:1000) |
| Antibody | Anti-HSP90 (C45G5) (Rabbit monoclonal) | Cell Signaling Technologies | Cat# 4877S RRID:AB_2233307 | WB (1:1000) |
| Antibody | Anti-Rheb (E1G1R) (Rabbit monoclonal) | Cell Signaling Technologies | Cat# 13879 RRID:AB_2721022 | WB (1:1000) IF (1:800) |
| Antibody | Anti-TSC2 (D93F12) (Rabbit monoclonal) | Cell Signaling Technologies | Cat# 4308 RRID:AB_10547134 | WB (1:1000) IF (1:100) |
| Antibody | Anti-Akt (Rabbit polyclonal) | Cell Signaling Technologies | Cat# 9272 RRID:AB_329827 | WB (1:1000) |
| Antibody | Anti-Deptor (Rabbit polyclonal) | Novus Biologicals | Cat# NBP1-49674 RRID:AB_10011798 | WB (1:1000) |
| Antibody | Anti-phospho-AMPKα (Thr172) (Rabbit monoclonal) | Cell Signaling Technologies | Cat# 2535 RRID:AB_331250 | WB (1:1000) |

*Continued on next page*

*Continued*

| Reagent type (species) or resource | Designation | Source or reference | Identifiers | Additional information |
|---|---|---|---|---|
| Antibody | Anti-Ki67 (D3B5) (Rabbit monoclonal) | Cell Signaling Technologies | Cat# 12202 RRID:AB_2620142 | IHC (1:400) |
| Antibody | Anti-phospho-S6 (Ser235/236) (Rabbit monoclonal) | Cell Signaling Technologies | Cat# 4856 RRID:AB_2181037 | WB (1:1000) |
| Antibody | Anti-S6 (54D2) (Mouse monoclonal) | Cell Signaling Technologies | Cat# 2317 RRID:AB_2238583 | WB (1:1000) |
| Antibody | Anti-Rheb (B-12) (Mouse monoclonal) | Santa Cruz Biotechnology | Cat# sc-271509 RRID:AB_10659102 | PLA (1:50) IP (1:20) |
| Antibody | Anti-phospho-4EBP1 (Ser65) (Rabbit polyclonal) | Cell Signaling Technologies | Cat# 9451 RRID:AB_330947 | WB (1:1000) |
| Antibody | Anti-4EBP1 (53H11) (Rabbit monoclonal) | Cell Signaling Technologies | Cat# 9644 RRID:AB_2097841 | WB (1:1000) |
| Antibody | Anti-p53 (1C12) (Mouse monoclonal) | Cell Signaling Technologies | Cat# 2524 RRID:AB_331743 | WB (1:1000) |
| Antibody | Anti-Sco2 (Rabbit polyclonal) | LS Bio | Cat# LS-C349015 | WB (1:500) |
| Antibody | Anti-LAMP1(H4A3) (Mouse monoclonal) | Santa Cruz Biotechnology | Cat# sc-20011 RRID:AB_626853 | IF (1:50) |
| Antibody | Alexa Fluor 594 AffiniPure Goat Anti-Rabbit IgG (H+L) | Jackson Immuno Research Laboratories | Cat #111-585-144 RRID:AB_2307325 | IF (1:100) |
| Antibody | Alexa Fluor 488 AffiniPure Goat Anti-Mouse IgG (H+L) | Jackson Immuno Research Laboratories | Cat #115-545-062 RRID:AB_2338845 | IF (1:100) |
| Antibody | Peroxidase AffiniPure F(ab')2 Goat Anti-Mouse IgG (H+L) | Jackson Immuno Research Laboratories | Cat #115-036-062 RRID:AB_2307346 | WB (1:10,000) |
| Antibody | Peroxidase AffiniPure F(ab')2 Goat Anti-Rabbit IgG (H+L) | Jackson Immuno Research Laboratories | Cat #111-036-003 RRID:AB_2337942 | WB (1:10,000) |
| Antibody | Peroxidase AffiniPure F(ab')2 Donkey Anti-Rabbit IgG (H+L) | Jackson Immuno Research Laboratories | Cat #711-036-152 RRID:AB_2340590 | WB (1:10,000) |
| Commercial assay or kit | Seahorse XF Cell Mito Stress Kit | Agilent | Part Number: 1037015–100 | |
| Commercial assay or kit | Seahorse XF Glycolytic Rate Assay | Agilent | Part Number: 103344–100 | |

*Continued on next page*

*Continued*

| Reagent type (species) or resource | Designation | Source or reference | Identifiers | Additional information |
|---|---|---|---|---|
| Commercial assay or kit | GSH/GSSG Glo Assay | Promega | Cat# V6611 | |
| Commercial assay or kit | PLA Duolink In Situ Starter Kit | Sigma Aldrich | DUO92101 | |
| Chemical compound, drug | Rapamycin | Sigma Aldrich | R0395; CAS 53123-88-9 | |
| Chemical compound, drug | Torin1 | Cayman Chemical | Item No. 10997; CAS 1222998-36-8 | |
| Chemical compound, drug | BMH (bismaleimi dohexane) | Thermo Fisher Scientific | Cat# 22330; CAS 4856-87-5 | |
| Chemical compound, drug | DEM (diethylmaleate) | Sigma Aldrich | D97703; CAS 141-05-9 | |
| Chemical compound, drug | BSO (buthionine sulfoximine) | Cayman Chemical | Item no. 14484; CAS 83730-53-4 | |
| Chemical compound, drug | Erastin | Cayman Chemical | Item no. 17754; CAS 571203-78-6 | |
| Chemical compound, drug | GSH (L-Glutathione) | Sigma Aldrich | Cat# G6013, CAS 70-18-8 | |
| Chemical compound, drug | L-Methionine | Sigma Aldrich | Cat# M9625, CAS 63-68-3 | |
| Chemical compound, drug | L-Glutamine | Thermo Fisher Scientific | Cat# 25030081 | |
| Chemical compound, drug | L-Cystine | Sigma Aldrich | Cat# C6727 | |
| Software, algorithm | ImageJ | NIH | RRID:SCR_003070 | |
| Software, algorithm | GraphPad Prism | GraphPad | RRID:SCR_002798 | |
| Other | Pyruvate-free DMEM | Thermo Fisher Scientific | Cat# 21013024 | |
| Other | Glucose-free DMEM | Thermo Fisher Scientific | Cat# 11966025 | |
| Other | EBSS | Thermo Fisher Scientific | Cat# 24010043 | |
| Other | MEM Vitamin | Thermo Fisher Scientific | Cat# 11120052 | |
| Other | MEM Amino Acids | Thermo Fisher Scientific | Cat# 1130051 | |

*Continued on next page*

*Continued*

| Reagent type (species) or resource | Designation | Source or reference | Identifiers | Additional information |
|---|---|---|---|---|
| Other | HBSS | Thermo Fisher Scientific | Cat# 14025092 | |
| Other | Trypan Blue | Thermo Fisher Scientific | Cat# 15250061 | |
| Other | Protein G Agarose | Thermo Fisher Scientific | Cat# 15920010 | |
| Other | MitoTracker Green | Thermo Fisher Scientific | Cat# M7514 | |

## Mammalian cell culture

All cell lines have been confirmed of identity using STR profiling; most were obtained by the Coriell Institute. All were confirmed to be free of mycoplasma prior to each experiment. WT and S47 MEFs were generated and maintained as previously described (*Jennis et al., 2016*). Human WT LCLs (Catalog ID GM18870) and S47 LCLs (Catalog ID GM18871) were obtained from the Coriell Institute (Camden, New Jersey) and maintained as previously described (*Jennis et al., 2016*). MEF cultured cells were grown in DMEM (Corning Cellgro) supplemented with 10% fetal bovine serum (HyClone, GE Healthcare Life Sciences) and 1% penicillin/streptomycin (Corning Cellgro). Human LCLs were grown in RPMI (Corning Cellgro) supplemented with 15% heat inactivated fetal bovine serum (HyClone, GE Healthcare Life Sciences) and 1% penicillin/streptomycin (Corning Cellgro). Cells were grown in a 5% $CO_2$ humidified incubator at 37°C. For serum starvation experiments, cells were starved in DMEM containing 0.1% FBS for 16 hr. Following starvation, DMEM containing 10% FBS was re-introduced and cells were harvested at 0 min, 10 min, 30 min, 1 hr, 2 hr, and 8 hr after this point. For glucose starvation experiments, cells were starved in glucose-free DMEM (Thermo Fisher Scientific 11966025) for 16 hr. Following starvation, DMEM containing 4.5 g/L glucose was re-introduced and cells were harvested at 0 min, 10 min, 30 min, 1 hr, 2 hr, and 5 hr after this point. For amino acid starvation experiments, cells were starved for 4 hr in EBSS (Thermo Fisher Scientific 24010043) containing 25 mM glucose, 0.5 mM Glutamine, 1X MEM Vitamin (Thermo Fisher Scientific 11120052), 0.2% FBS, 25 mM HEPES, 1X Penicillin/Streptomycin. Following starvation, the same media recipe now containing 1X MEM Amino Acids (Thermo Fisher 1130051) was re-introduced and cells were harvested at 0 min, 10 min, 30 min, 1 hr, 2 hr, and 5 hr after this point. For HBSS experiments, cells were washed once with PBS (Corning 21–031-CV) and then incubated with HBSS (Thermo Fisher Scientific 14025092) for 0, 2, or 6 hr. Viability was assessed using Trypan Blue (Thermo Fisher Scientific 15250061).

## Western blot

For western blot analyses, 50–100 µg of protein was resolved over SDS-PAGE gels using 10% NuPAGE Bis-Tris precast gels (Life Technologies) and were then transferred onto polyvinylidene difluoride membranes (IPVH00010, pore size: 0.45 mm; Millipore Sigma). Membranes were blocked for 1 hr in 5% bovine albumin serum (Sigma Aldrich, A9647). The following antibodies were used for western blot analyses: phospho-mTOR 1:1000 (Cell Signaling, 2971), mTOR 1:1000 (7C10, Cell Signaling, 2983), phospho-p70S6K1 1:1000 (Cell Signaling, 9205), p70S6K1 1:1000 (Cell Signaling, 9202), GAPDH 1:10,000 (14C10, Cell Signaling, 2118), TFAM 1:2000 (Abcam, ab131607), MTCO1 1:2000 (Abcam, ab14705), SDHA 1:1000 (Cell Signaling, 5839), Tom20 1:100 (F-10, Santa Cruz, sc17764), phospho-Akt (D9E, Cell Signaling, 4060), p62 1:1000 (Cell Signaling, 5114), LC3B 1:1000 (D11, Cell Signaling, 3868), HSP90 1:1000 (Cell Signaling, 4877S), Rheb 1:1000 (E1G1R, Cell Signaling, 13879), TSC2 1:1000 (D93F12, Cell Signaling, 4308), Akt 1:1000 (Cell Signaling, 9272), Deptor 1:1000 (Novus Bio, NBP1-49674SS), phospho-AMPKα (Cell Signaling, 2535). Rabbit or mouse secondary antibodies conjugated to horseradish peroxidase were used at a 1:10,000 dilution (Jackson

Immunochemicals), followed by a 5-min treatment with ECL (Amersham, RPN2232). Protein levels were detected using autoradiography and densitometry analysis of protein content was conducted using ImageJ software (NIH, Rockville, MD).

## Immunohistochemistry

Tissues were harvested and fixed in formalin overnight at 4°C, followed by a wash with 1X PBS and were then placed in 70% ethanol prior to paraffin embedding. The Wistar Institute Histotechnology Facility performed the tissue embedding and sectioning. For the immunohistochemistry (IHC) studies, paraffin embedded tissue sections were de-paraffinized in xylene (Fisher, X5-SK4) and re-hydrated in ethanol (100%–95%-85–75%) followed by distilled water. Samples underwent antigen retrieval by steaming slides in 10 mM Citrate Buffer (pH 6). Endogenous peroxidase activity was quenched with 3% hydrogen peroxide and slides were incubated in blocking buffer (Vector Laboratories, S-2012) for 1 hr. The slides were incubated with phospho-p70S6K1 (1:100, ThermoFisher Scientific, PA5-37733) or phospho-mTOR (1:100, Cell Signaling, 2971) primary antibody overnight at 4°C. The following day, slides were washed with PBS and incubated with HRP-conjugated secondary antibody for 30 min. Antibody complexes were detected using DAB chromogen (D5637). Light counterstaining was done with hematoxylin. Slides were imaged using the Nikon 80i upright microscope and at least four fields were taken per section.

## Co-immunoprecipitation

Following overnight seeding of WT and S47 immortalized MEFs, the cells were washed with 1X DPBS and the cell culture medium was replaced with 1% FBS DMEM medium [pyruvate-free DMEM (Thermo Fisher Scientific #21013024) supplemented with 1% FBS, 1% penicillin/streptomycin, 0.1 mM L-Methionine, 0.5 mM L-Glutamine, and 0.033 mM L-Cystine]. WT cells were treated with PBS or 3 mM GSH for 24 hr; while the S47 cells were treated with PBS for 5 hr or 24 hr, 100 µM BSO for 24 hr, or 50 µM DEM for 5 hr. Cells were harvested and centrifuged at 500 x g for 10 min at 4°C. The cell pellets were resuspended in CHAPS Lysis Buffer (1X DPBS with 0.3% CHAPS and freshly added protease inhibitors) at 4°C. Cell disruption was performed by passing the cells through a 23-gauge needle attached to a 1 ml syringe. The skeletal muscles were homogenized using the Qiagen Tissue Lyser II. Total cellular homogenates were rotated/nutated at 4°C for 30 min, and spun at 11,000 x g for 20 min at 4°C. Protein extracts (3 mg per reaction) were incubated with the Rheb antibody (Santa Cruz, sc-271509) overnight at 4°C. The Rheb-immunocomplexes were captured using recombinant protein G agarose (Thermo Fisher Scientific, 15920010) at 4°C for 2 hr. Resins were washed three times using the CHAPS Lysis Buffer. Equal volumes of 2x Laemmli Sample Buffer were added to each reaction, samples were heated for 10 min at 100°C. The Rheb-associated proteins were analyzed by western blotting, using GAPDH (Cell Signaling, 2118), Rheb (Cell Signaling, 13879), and mTOR (7C10, Cell Signaling, 2983) antibodies.

## Mitochondrial metabolism and mTOR inhibition assays

The oxygen consumption rate (OCR) and glycolytic rate were determined using the Seahorse XF MitoStress Assay and the Seahorse XF Glycolytic Rate Assay, respectively, according to the manufacturer's protocol. Cells were plated one day prior to the assay, LCLs at 100,000 cells/well and MEFs at 60,000 cells/well. LCLs were treated with 200 nM rapamycin, 100 nM Torin1 or 1 µM Torin1 for 24 hr prior to running the MitoStress Assay. To assess differences in mTOR inhibition, WT and S47 LCLs were treated with 200 nM rapamycin or 100 nM Torin1 for 0 min, 10 min, 30 min, 1 hr, 2 hr, or 6 hr and subsequently cells were harvested for western blot analysis. To determine mitochondrial content, WT and S47 MEFs were incubated with 500 nM of MitoTracker Green (ThermoFisher Scientific, M7514) for 1 hr at 37°C. Cells were then spun down, washed once with PBS, spun down and resuspended in PBS. The FACSCelesta (BD Biosciences) was used to detect fluorescence and at least 10,000 events were measured per sample.

## Metabolite measurements

Media was collected after 24 hr after plating LCLs or MEFs, and the YSI-71000 Bioanalyzer was used to determine glucose, glutamine, lactate and glutamate levels as previously described (*Londoño Gentile et al., 2013*). For the metabolic flux studies, cells were incubated in uniformly

labeled $^{13}$C-glucose (25 mM) as indicated in the figure legends. For intracellular extracts, after incubation, the culture medium was aspirated and cells were washed once in ice-cold PBS. Metabolites were extracted by adding a solution of methanol/acetonitrile/water (5:3:2) to the well. Plates were incubated at 4°C for 5 min on a rocker and then the extraction solution was collected. The metabolite extract was cleared by centrifuging at 15,000 x *g* for 10 min at 4°C. Supernatants were transferred to LC-MS silanized glass vials with PTFE caps and either run immediately on the LC-MS or stored at −80°C. LC-MS analysis was performed on a Q Exactive Hybrid Quadrupole-Orbitrap HF-X MS (ThermoFisher Scientific) equipped with a HESI II probe and coupled to a Vanquish Horizon UHPLC system (ThermoFisher Scientific). 0.002 ml of sample is injected and separated by HILIC chromatography on a ZIC-pHILIC 2.1 mm. Samples were separated by ammonium carbonate, 0.1% ammonium hydroxide, pH 9.2, and mobile phase B is acetonitrile. The LC was run at a flow rate of 0.2 ml/min and the gradient used was as follows: 0 min, 85% B; 2 min, 85% B; 17 min, 20% B; 17.1 min, 85% B; and 26 min, 85% B. The column was maintained at 45°C and the mobile phase was also pre-heated at 45°C before flowing into the column. The relevant MS parameters were as listed: sheath gas, 40; auxiliary gas, 10; sweep gas, 1; auxiliary gas heater temperature, 350°C; spray voltage, 3.5 kV for the positive mode and 3.2 kV for the negative mode. Capillary temperature was set at 325°C, and funnel RF level at 40. Samples were analyzed in full MS scan with polarity switching at scan range 65 to 975 m/z; 120,000 resolution; automated gain control (AGC) target of 1E6; and maximum injection time (max IT) of 100 milliseconds. Identification and quantitation of metabolites was performed using an annotated compound library and TraceFinder 4.1 software. The 'M+X' nomenclature refers to the isotopologue for that given metabolite. Isotopologues are chemically identical metabolites that differ only in their number of carbon-13 atoms. For instance, 'M+two citrate' means that two of the six carbons in citrate are carbon-13 while the other four are carbon-12. 'M+four citrate' means that four of the six carbons in citrate are carbon-13 while the other two are carbon-12. Isotopologue fractional labeling was corrected for carbon-13 natural abundance.

## GSH/GSSG abundance and BMH crosslinking

Relative GSH/GSSG abundance was measured using the GSH/GSSG-Glo Assay (Promega catalog #V6611), according to the manufacturer's instruction. Immortalized WT and S47 MEFs were generated and maintained as previously described (*Jennis et al., 2016*; *Leu et al., 2019*). For BMH crosslinking studies, the WT and S47 cells were cultured in 1% FBS DMEM medium and treated with PBS or 50 µM diethyl maleate (DEM, ThermoFisher Scientific AC114440010) for 5 hr; PBS or 100 µM BSO (Cayman Chemical item #14484) for 24 hr; or DMSO or 2 µM Erastin (Cayman Chemical item #17754) for 24 hr. Proteins were extracted from cultured cells or mouse tissue (skeletal muscle, lungs) using 1X DPBS (Thermo Fisher Scientific 14190144) supplemented with 0.5% IGEPAL CA-630, 1 mM PMSF, 6 µg/ml aprotinin, and 6 µg/ml leupeptin at 4 °C. The tissues were homogenized using the Wheaton Overhead Stirrer. Total cellular homogenates were pulse sonicated using the Branson digital sonifier set at 39% amplitude. Total protein extracts (100 µg per reaction) were incubated with or without 1 mM BMH (Thermo Fisher Scientific 22330) for 30 min at 30°C. The samples were quenched with an equal volume of 2x Laemmli Sample Buffer (BioRad 1610737) supplemented with 5% β-Mercaptoethanol (BioRad 1610710) and heated for 10 min at 100°C. The protein samples were size fractionated on Novex 4–20% Tris-Glycine Mini Gels (Thermo Fisher Scientific XP04200BOX) at room temperature and subsequently transferred overnight onto Immuno-Blot PVDF membranes (BioRad 1620177) at 4°C. The membranes were blocked with 3% nonfat dry milk (BioRad 1706404) in 1X PBST for 30 min at room temperature and incubated with the GAPDH antibody (Cell Signaling Technology 2118) overnight with rotation/nutation at 4°C. After washing the blots in 1X PBST, the membranes were incubated with Donkey anti-Rabbit (Jackson ImmunoResearch 711-036-152) for 2 hr at room temperature. Membrane-immobilized protein detection used ECL western blotting detection reagents (GE Healthcare RPN2106; Millipore Sigma GERPN2106).

## Proximity ligation assay

Cells were grown on Lab-Tek II eight-well chamber slides, and were either untreated, treated with 50 µM diethyl maleate (DEM, ThermoFisher Scientific AC114440010) for 5 hr or treated with 10 µM of buthionine sulfoximine for 24 hr (BSO, Cayman Chemicals, 83730-53-4) and fixed with 4% paraformaldehyde (Electron Microscopy Sciences, 15710). Protein-protein interactions were assessed using

the PLA Duolink in situ starter kit (Sigma Aldrich, DUO92101) following the manufacturer's protocol. The following primary antibodies were used: Rheb 1:50 (B-12, Santa Cruz, sc271509), mTOR 1:500 (7C10, Cell Signaling, 2983), GAPDH 1:1000 (14C10, Cell Signaling, 2118). ImageJ software (NIH, Rockville, MD) was used to quantify PLA signals.

## Immunofluorescence staining

Cells were fixed in 4% paraformaldehyde for 10 min, followed by 3 PBS washes and then permeabilization with 0.25% Triton X-100 for 10 min. Cells were washed 3x with PBS, blocked for 1 hr in a PBS solution containing 1% bovine serum albumin and 5% normal goat serum (Jackson Immunoresearch 005-000-121). Cells were incubated overnight at 4°C with the following primary antibodies diluted in blocking buffer: Rheb 1:800 (Cell Signaling Technologies, #13879), TSC2 1:100 (Cell Signaling Technologies, #4308), LAMP1 1:50 (Santa Cruz, sc-20011). Cells were washed with PBS and incubated with the following secondary antibodies at 37°C for 45 min: Alexa Fluor 594 AffiniPure Goat Anti-Rabbit IgG (Jackson Immunoresearch 111-585-144) and Alexa Fluor 488 AffiniPure Goat Anti-Mouse IgG (Jackson Immunoresearch 115-545-062). The cells were mounted with media containing DAPI and images were captured using the Leica TSC SP5 microscope.

## Body composition and metabolic cage studies

WT and S47 mice in a pure C57Bl/6 background are previously described (*Jennis et al., 2016*). All mouse studies were performed in accordance with the guidelines in the Guide for the Care and Use of Laboratory Animals of the NIH and all protocols were approved by the Wistar Institute Institutional Animal Care and Use Committee (IACUC). Mice were fed an ad libitum diet and were housed in plastic cages with a 12 hr/12 hr light cycle at 22°C unless otherwise stated. Fat and lean content were measured in live male mice at 6 weeks of age using nuclear magnetic resonance (NMR) with the Minispec LF90 (Bruker Biospin, Billerica, MA). Indirect calorimetry was conducted to assess metabolic capabilities in mice (Oxyman/Comprehensive Laboratory Animal Monitoring System (CLAMS); Columbus Instruments). Data for the WT mice was previously published in supplemental data of *Kung et al., 2016*. Six-week-old mice were single caged, provided with water and food ad libitum and allowed to acclimate to the cages for 2 days. Oxygen consumption ($VO_2$) and carbon dioxide production ($VCO_2$) were recorded for 48 hr using an air flow of 600 ml/min and temperature of 22°C. Respiratory exchange ratio (RER) is calculated as $VCO_2/VO_2$ and heat (kcal/h) is calculated by 3.815 + 1.232*(RER). Photodetectors were used to measure physical activity (Optovarimex System; Columbus Instruments).

## Treadmill and serum metabolite studies

Mice were allowed to acclimate to the metabolic treadmill (Columbus Instruments) for 5 min before beginning their runs. The treadmill was then set to 5 m/min and speed increased by 5 m/min every 2 min until 20 m/min was reached. Upon reaching 20 m/min, the incline was increased by 5° every 2 min until reaching a maximum of 25 degrees. Mice were allowed to run at this maximum speed and incline until exhaustion, defined by the mice spending 10 continuous seconds on the shock grid. Lactate (Nova Biomedical) and glucose (One Touch) measurements were taken using test strips just prior to treadmill entry and immediately after exhaustion using handheld meters. Tail blood was also taken prior to treadmill entry and immediately after exhaustion and metabolites measured using the Vettest serum analyzer (Idexx Laboratories).

## Statistical analysis

Unless otherwise stated, all experiments were performed in triplicate. The two-tailed unpaired Student t-test was performed. All in vitro data are reported as the mean ± SD unless stated otherwise, and in vivo are reported as the mean ± SE. Statistical analyses were performed using GraphPad Prism, p-values are as follows: (*) p-value<0.05, (**) p-value<0.01, (***) p-value<0.001, (****) p-value<0.0001. For the CLAMs and mouse exercise data, the Wilcoxon rank-sum test was used to compare the differences between S47 and WT mice.

## Acknowledgements

This work was supported by R01 CA102184 (MEM), R01 CA139319 (DLG and MEM), R01 CA238611 (MEM), P01 CA114046 (DLG and MEM), T32 CA009071, F32 CA220972 and K99 CA241367 (TB), R01 CA174761 (KEW), R01 AG043483 and DK098656 (JAB) and the Rodent Metabolic Phenotyping Core (P30-DK19525), and DP2-CA249951 (ZTS). RSA is partly supported by a Bloomberg Distinguished Professorship. The authors acknowledge the Histotechnology, Laboratory Animal and Imaging facilities at the Wistar Institute. The authors are grateful to Allie Lipshutz and Lindsey Schweitzer for expert technical help, and Matthew Jennis for the mouse weight data.

## Additional information

### Competing interests

Maureen E Murphy: Senior Editor, eLife. The other authors declare that no competing interests exist.

### Funding

| Funder | Grant reference number | Author |
|---|---|---|
| National Cancer Institute | R01 CA102184 | Maureen E Murphy |
| National Cancer Institute | R01 CA139319 | Donna L George<br>Maureen E Murphy |
| National Cancer Institute | CA238611 | Maureen E Murphy |
| National Cancer Institute | P01 CA114046 | Donna L George<br>Maureen E Murphy |
| National Cancer Institute | T32 CA009171 | Thibaut Barnoud |
| National Cancer Institute | F32 CA220972 | Thibaut Barnoud |
| National Cancer Institute | K99 CA241367 | Thibaut Barnoud |
| National Cancer Institute | R01 CA174761 | Kathryn E Wellen |
| National Institute on Aging | R01 AG043483 | Joseph A Baur |
| National Institute of Diabetes and Digestive and Kidney Diseases | R01 DK098656 | Joseph A Baur |
| National Institute of Diabetes and Digestive and Kidney Diseases | P30 DK19525 | Joseph A Baur |
| National Cancer Institute | DP2-CA249951 | Zachary T Schug |
| Bloomberg Family Foundation | | Rexford Ahima |
| National Cancer Institute | Cancer Center Support Grant (CCSG) P30 CA010815 | Maureen E Murphy |

The funders had no role in study design, data collection and interpretation, or the decision to submit the work for publication.

### Author contributions

Keerthana Gnanapradeepan, Conceptualization, Formal analysis, Validation, Investigation, Visualization, Methodology, Writing - original draft, Writing - review and editing; Julia I-Ju Leu, Conceptualization, Formal analysis, Investigation, Methodology, Writing - review and editing; Subhasree Basu, Conceptualization, Formal analysis, Investigation, Writing - review and editing; Thibaut Barnoud, Conceptualization, Formal analysis, Funding acquisition, Investigation, Writing - review and editing; Madeline Good, Formal analysis, Investigation; Joyce V Lee, William J Quinn, Che-Pei Kung, Data curation, Formal analysis, Investigation, Writing - review and editing; Rexford Ahima, Joseph A Baur, Kathryn E Wellen, Data curation, Funding acquisition, Methodology, Writing - review and editing;

Qin Liu, Formal analysis, Methodology, Writing - review and editing; Zachary T Schug, Data curation, Formal analysis, Funding acquisition, Investigation, Methodology, Writing - review and editing; Donna L George, Funding acquisition, Methodology, Writing - review and editing; Maureen E Murphy, Conceptualization, Formal analysis, Funding acquisition, Writing - review and editing

### Author ORCIDs
Keerthana Gnanapradeepan (iD) https://orcid.org/0000-0002-8984-9742
Thibaut Barnoud (iD) http://orcid.org/0000-0001-5588-6281
Che-Pei Kung (iD) http://orcid.org/0000-0002-1150-4998
Joseph A Baur (iD) http://orcid.org/0000-0001-8262-6549
Kathryn E Wellen (iD) https://orcid.org/0000-0002-2281-0042
Maureen E Murphy (iD) https://orcid.org/0000-0001-7644-7296

### Ethics
Animal experimentation: This study was performed in strict accordance with the recommendations in the Guide for the Care and Use of Laboratory Animals of the National Institutes of Health. All of the animals were handled according to approved institutional animal care and use committee (IACUC) protocols of the University of Pennsylvania (protocol 804474) and The Wistar Institute (protocol 201283).

### Decision letter and Author response
Decision letter https://doi.org/10.7554/eLife.55994.sa1
Author response https://doi.org/10.7554/eLife.55994.sa2

## Additional files

### Supplementary files
• Transparent reporting form

### Data availability
All data generated or analysed during this study are included in the manuscript and supporting files. Source data files are provided for Figure 6 and Figure 2—figure supplement 1.

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
