## [Decision Letter]

**Acceptance summary:**

This work highlights a novel mechanism whereby the p53 tumor suppressor protein regulates mTOR activity, via the control of cellular redox state. Being based on using both mouse and human tissues and cells containing an African-centric variant of p53, the work has direct implications for understanding cancer disparities in African-descent populations. This work suggests that enhanced mTOR activity and metabolic efficiency may have been selected for in early Africa, making this variant one of a growing number of cancer-predisposing variants.

**Decision letter after peer review:**

[Editors’ note: the authors submitted for reconsideration following the decision after peer review. What follows is the decision letter after the first round of review.]

Thank you for submitting your work entitled "Increased mTOR activity and metabolic efficiency in mice and human cells containing the tumor-prone p53 variant Pro47Ser" for consideration by *eLife*. Your article has been reviewed by three peer reviewers, one of whom is a member of our Board of Reviewing Editors, and the evaluation has been overseen by a Reviewing Editor and a Senior Editor. The following individuals involved in review of your submission have agreed to reveal their identity: Gina DeNicola (Reviewer #2).

This manuscript was reviewed by three experts in the field. While all three reviewers agreed that the findings were of considerable potential interest, all reviewers agreed that the manuscript in its current form did not reach the level of mechanistic depth required for publication in *eLife*. The reviewer consensus was that considerable additional experiments would be required and that these added experiments could not realistically be done in two months.

Specifically, the reviewers noted two major areas which would require additional experimentation. First, the link between the S47 variant of p53, GSH levels and mTOR activity required additional experiments and orthogonal approaches to validate the effect of p53 P47S on GSH levels and to connect altered GSH to mTORC1 activation. Second, further experiments are required to assess the degree to which the reported metabolic phenotypes arise as a consequence of increased mTORC1 activity. Given the length of time required to perform the recommended experiments, we are returning the manuscript and reviews to you so that they may guide you as you seek publication elsewhere. Should you wish to resubmit to *eLife* at a future date, we would be happy to reconsider a resubmission with increased mechanistic detail.

Reviewer #1:

In this manuscript, the authors present a metabolic characterization of cells and mice expressing the S47 variant of the tumor suppressor p53. S47 lines have increased basal mTOR activity, increased nutrient consumption and increased capacity for oxygen consumption. Mechanistically, the increase in mTOR activity is linked to alterations in GSH levels. in vivo, mice expressing S47 are bigger but do not consume more food or exhibit more activity. Nevertheless, these mice demonstrate increased capacity to respond to an exercise challenge. While the manuscript reports several interesting observations, how the various phenotypes relate to each other is unclear. While the link between p53, redox stress, and mTOR activity via alterations in GAPDH levels is a very interesting new mode of metabolic control by p53, several important experiments are required to strengthen this link.

Essential revisions:

1) The link between GSH levels and mTOR activity requires further confirmation. What do manipulations to intracellular GSH do to mTOR activity in S47 cells? Is increasing GSH, for example adding eGSH, sufficient to activate mTOR in WT cells? Is activation of mTOR downstream of altered cysteine regulation by the S47 variant? Would adding cell-permeable reduced cysteine in the form of N-acetylcysteine restore mTOR activity back to normal levels? Additionally, orthogonal validation of the changes in Rheb interactions following GSH manipulation, for example by co-IP, would further strengthen this point.

2) To what extent whole-body metabolic phenotypes are related to changes in mTOR activity is not addressed. In some cases, the phenotypes are not consistent with genetic manipulation of S6K, which increases organismal oxygen consumption and exercise capacity. There are many potential reasons why mice might perform differently in response to an exercise challenge: for example, altered muscle fiber type composition, among others. Furthermore, how the in vitro metabolic assays, showing increased nutrient consumption and increased oxygen consumption, relate to the in vivo phenotypes is not clear.

Reviewer #2:

In their manuscript "Increased mTOR activity and metabolic efficiency in mice and human cells containing the tumor predisposing p53 variant Pro47Ser", Gnanapradeepan et al., describe an association between this variant and increased mTOR activity, oxidative mitochondrial metabolism, and metabolic efficiency. This finding is interesting and may serve to explain how this variant, found in sub-Saharan African populations, may confer fitness at the organismal level while concomitantly increasing cancer risk. Mechanistically, they propose that increased glutathione content in P47S cells promotes the association between mTOR and Rheb via an alteration in the redox state of GAPDH. However, this mechanism needs further development.

Essential revisions:

1) DEM is a non-specific alkylator of thiols, including those found in proteins. It is therefore not surprising that DEM would influence the mobility in BMH-treated samples. The authors should repeat this experiment with BSO. Further, as the authors have proposed SLC7A11 as the mediation of redox/GSH modulator, does erastin treatment produce the same effect?

2) The authors have not demonstrated a causal role for the altered redox state of GAPDH in the association between mTOR and Rheb. Can exogenous GSH (ester) and/or SLC7A11 overexpression influence mTOR signaling in WT cells? Can a GAPDH mutant that is insensitive to oxidation (C152S) influence the mTOR-Rheb interaction?

3) The authors do not show evidence that cysteine and GSH in are higher in S47 cells. Rather, they reference their previous study. When using compounds like DEM and BSO, evidence for efficacy in their current experiments is necessary. Further, GSH levels in mouse tissues should be included to support the link between mTOR signaling and GSH. Do only the mouse tissues with altered mTOR signaling have altered GSH?

Reviewer #3:

Gnanapradeepan et al., set out to dissect the metabolic outputs of a p53 variant (S47), present in African-descent populations and associated with increased cancer risk. Their previous work on the S47 variant affecting several negative regulators of mTORC1 signaling prompted them to examine mTORC1 signaling. Using both cell culture and their previously-reported S47 mouse model, the authors observe enhanced phosphorylation of mTORC1 substrate p-S6K1 in S47 cells and tissues, but no detectable changes in the phosphorylation of 4EBP1, another key substrate of mTORC1. Mechanistically, they attribute the enhanced mTORC1 activity to an increased mTOR-Rheb interaction, at the expense of Rheb-GAPDH interaction.

The authors then set out to determine the functional consequence of mTORC1 activity in metabolism and find several interesting metabolic phenotypes associated with the S47 variant, including increased glycolytic rate and OCR, enhanced muscle and body mass, and improved metabolic efficiency.

While some of the findings are interesting, it is unclear whether the changes in mTORC1 activity are connected to the observed metabolic phenotypes. Surprisingly, the authors have overlooked the well-known primary modes of regulation of mTORC1 activity (e.g., mTOR/Rheb localization at the lysosome, TSC phosphorylation, and AMPK signaling), but instead focus on the obscure GAPDH-Rheb-mTOR interaction, the contribution of which to mTORC1 activation is unclear and only reported under low-glucose conditions. More work is necessary to substantiate the authors' conclusions.

Essential revisions:

1) The metabolic phenotypes that the authors associate with mTORC1 signaling are based on correlative observations. The authors should attempt to rescue some of the observed phenotypes in Figure 2D, 2E, and Figure 6 using both allosteric (Rapamycin) and catalytic (Torin-1) inhibitors of mTORC1.

2) Figure 3: What are the authors' thoughts on Rapamycin failing to decrease OCR in S47 cells to the same extent as in WT cells? Is this due to partial inhibition of mTORC1 activity in these cells? Are the FKBP12 levels similar in these cells? Is ECAR dependent on mTORC1? The authors should also use a catalytic inhibitor of mTORC1 (e.g., Torin-1) and compare its effect to the effect of Rapamycin. It would be necessary to determine inhibition of mTORC1 signaling by these inhibitors on multiple substrates of mTORC1 (at least, S6K1 and 4EBP1) to be able to draw conclusions on this.

3) Figure 5: More work is needed to substantiate the mechanistic conclusions on increased mTORC1 activity in S47 cells because of an enhanced mTORC1-Rheb association and reduced GAPDH-Rheb. Using Proximity Ligation Assay (PLA) assays, the authors suggest that the Rheb-GAPDH association is dependent on the redox status of GAPDH, and in a reduced state as in S47, the GAPDH-Rheb interaction would decrease, allowing for more Rheb to associate with mTOR. While this is an interesting hypothesis, the authors should first validate the reagents used in their studies and employ orthogonal methods to verify these findings. The antibodies used for the PLA should be validated using knockdown experiments followed by PLA. Complementary experiments such as immunoprecipitation assays should be used to test the mTOR-Rheb and GAPDH-Rheb localization interaction changes in WT vs. S47 contexts.

4) Since the primary mode of activation of mTORC1 is at the lysosomes, the authors should test whether mTORC1, Rheb and TSC2 localization on the lysosomes is affected or not in the S47 cells. No conclusions regarding overactivation of mTORC1 in S47 cells can be made without this crucial bit of data.

5) Figure 2F, G and Figure 2—figure supplement 1A, B: The authors conclude a higher flux of glucose carbon into the TCA cycle in S47 cells. Why do the authors think that these minute changes (the differences appear to be less than 5%) are biologically significant? Can the authors observe significant changes in glycolytic intermediates in the 13C6-Glucose tracing experiment? Could the changes in OCR be due to an effect on NAD/NADH instead?

[Editors' note: further revisions were suggested prior to acceptance, as described below.]

Thank you for choosing to send your work entitled "Increased mTOR activity and metabolic efficiency in mice and human cells containing the tumor-prone p53 variant Pro47Ser" for consideration at *eLife*. Your letter of appeal has been considered by a Senior Editor along with a Reviewing Editor and the original manuscript reviewers, and we are prepared to consider a revised submission with no guarantees of acceptance.

All three reviewers agreed that the initial submission required extensive additional data to support the connection between p53 S47 and the GSH-GAPDH-Rheb-mTORC1 axis. Additionally, the reviewers believed while the in vivo data were quite interesting for the potential to provide an evolutionary explanation for the persistence of the p53 S47 polymorphism, these findings lacked mechanistic connection to the mTORC1 phenotypes reported in vitro.

All reviewers have now reviewed your outline for revision experiments and agree that these proposed experiments will likely go a long way to strengthening the link between p53 S47, GSH, and mTORC1 activity in vitro. However, because the GAPDH-Rheb-mTORC1 relationship is not a well-established mechanism for mTORC1 activation in the field (to date, there is only one report supporting this mechanism, which is proposed to occur under glucose deprived conditions), the authors will need to perform experiments both to address this hypothesis and rule out other, more canonical explanations for regulation of mTORC1 activity. One reviewer emphasized that experiments testing the effects of mTORC1 activation in response to amino acids, insulin, or glucose in WT vs S47 cells are needed to clarify the mechanism of regulation of mTORC1 (related to Figure 1F).

While all reviewers appreciated that in vivo experiments probing the relevance of mTORC1 activity to the in vivo phenotypes are challenging, in the absence of such experiments the in vivo metabolic phenotypes cannot be directly linked to mTORC1 activation. One reviewer suggested that ex vivo experiments could be used to address this point. For example, the authors could perform a Seahorse assay on muscle fibers, for which protocols from Agilent are available, and/or PLA/western blots after modulation of glutathione. Lacking such experiments, the animal phenotypes remain correlative and disconnected from the cell culture experiments, and the authors would need to revise the manuscript to ensure that the conclusions drawn from the in vitro data were separated from presentation of the in vivo data.

We will look forward to receiving a revised article and a file describing the changes made in response to the decision and review comments for further consideration. Of course, given the current circumstances, we do not expect a revised manuscript within two months. Instead we give you as much time as you need to submit a revised submission. Please be aware that the revision requirements outlined above, along with your response to these comments, will be published should your article be accepted for publication, subject to author approval. In this event you acknowledge and agree that the these will be published under the terms of the Creative Commons Attribution license.

---

## [Author Response]

[Editors’ note: The authors appealed the original decision. What follows is the authors’ response to the first round of review.]

Reviewer #1:Essential revisions:1) The link between GSH levels and mTOR activity requires further confirmation. What do manipulations to intracellular GSH do to mTOR activity in S47 cells? Is increasing GSH, for example adding eGSH, sufficient to activate mTOR in WT cells? Is activation of mTOR downstream of altered cysteine regulation by the S47 variant? Would adding cell-permeable reduced cysteine in the form of N-acetylcysteine restore mTOR activity back to normal levels? Additionally, orthogonal validation of the changes in Rheb interactions following GSH manipulation, for example by co-IP, would further strengthen this point.

We thank the reviewer for these suggestions. We have extensively revised the figures to address these concerns. Specifically, we now show:

a) GSH levels clearly regulate mTOR activity. Specifically, five hour treatment of cells with the GSH alkylator DEM, and more dramatically the combination of DEM with glutamate (inhibits cystine import), causes a significant decrease in mTOR activity in S47 cells (new Figure 5B).

b) Moreover, by IP-western we now show: exogenous GSH decreases the Rheb-GAPDH complex in WT cells, while conversely, either DEM or BSO (the latter decreases GSH biosynthesis) increases the Rheb-GAPDH complex in S47 cells. These treatments do not affect the steady state level of Rheb or GAPDH (New Figure 5D). We hope the reviewer agrees that these new data strengthen our conclusions.

2) To what extent whole-body metabolic phenotypes are related to changes in mTOR activity is not addressed. In some cases, the phenotypes are not consistent with genetic manipulation of S6K, which increases organismal oxygen consumption and exercise capacity. There are many potential reasons why mice might perform differently in response to an exercise challenge: for example, altered muscle fiber type composition, among others. Furthermore, how the in vitro metabolic assays, showing increased nutrient consumption and increased oxygen consumption, relate to the in vivo phenotypes is not clear.

As discussed and agreed upon in the previously approved plan of attack, we have added a paragraph in the Discussion section that addresses this topic:

“One caveat of this study, however, is that we do not directly demonstrate that the increased mTOR activity in S47 mice is causing their increased lean content or superior performance on treadmill assays. Transient treatment with mTOR inhibitors elicits highly complex and often contrasting effects on energy expenditure and treadmill performance, likely due to the existence of feedback loops and the effect of inhibitors on multiple organ systems in the mouse. As just two examples: rapamycin has shown contrasting effects on energy expenditure in animals, depending on how long mice are treated (Fang et al., 2031); similarly, treatment of mice with rapamycin has shown limited impact on treadmill endurance, despite causing decreased expression of genes involved in mitochondrial biogenesis and oxidative phosphorylation in the muscle (Ye et al., 2013). Possibly the most consistent findings in the literature reflect the general consensus that mTORC1 is involved in mechanisms that drive increased muscle mass (Goodman, 2019) and that heightened mTOR activity leads to enhanced muscle recovery after exercise (Song er al., 2017; Yoon, 2017). It remains to be tested if these are the pathways affected in S47 mice.”

Reviewer #2:Essential revisions:1) DEM is a non-specific alkylator of thiols, including those found in proteins. It is therefore not surprising that DEM would influence the mobility in BMH-treated samples. The authors should repeat this experiment with BSO. Further, as the authors have proposed SLC7A11 as the mediation of redox/GSH modulator, does erastin treatment produce the same effect?

These are excellent suggestions, and we have done them. We now show that pre-treatment of cells with BSO for 24 hours also causes the mobility of BMH-cross-linked GAPDH in S47 cells to ‘restore’ to the mobility in WT cells (new Figure 5—figure supplement 1A). Likewise, we show that Erastin treatment causes GAPDH mobility in WT and S47 cells to shift toward identical mobility (new Figure 5—figure supplement 1B).

2) The authors have not demonstrated a causal role for the altered redox state of GAPDH in the association between mTOR and Rheb. Can exogenous GSH (ester) and/or SLC7A11 overexpression influence mTOR signaling in WT cells? Can a GAPDH mutant that is insensitive to oxidation (C152S) influence the mTOR-Rheb interaction?

This has been done. As indicated above (see similar comment 1 from reviewer 1), we now show that DEM (inactivates GSH), or a combination of DEM and glutamate (inhibits cystine import from the SLC7A11 transporter) markedly inhibits mTOR activity in S47 cells (new Figure 5B). We also show by IP-western that exogenous GSH decreases the Rheb-GAPDH complex in WT cells, while conversely, either DEM or BSO (the latter decreases GSH biosynthesis) increases the Rheb-GAPDH complex in S47 cells, while these treatments do not affect the steady state level of Rheb or GAPDH. (New Figure 5D). We thank the reviewer for these suggestions.

3) The authors do not show evidence that cysteine and GSH in are higher in S47 cells. Rather, they reference their previous study. When using compounds like DEM and BSO, evidence for efficacy in their current experiments is necessary. Further, GSH levels in mouse tissues should be included to support the link between mTOR signaling and GSH. Do only the mouse tissues with altered mTOR signaling have altered GSH?

These are excellent points, and this has been done. In new Figure 5A, we now show increased GSH:GSSG ratio in lung and skeletal muscle of S47 mice, compared to WT, and in S47 MEFs compared to WT. Further, in this figure, we show that the DEM successfully decreases these GSH:GSSG ratios in WT and S47 MEFs (new Figure 5A). With regard to the question “Do only the mouse tissues with altered mTOR signaling have altered GSH?”, we too wonder about this point and while our data in liver suggest this may be the case, we think it is premature to come to that conclusion. Therefore we state in the Discussion section that this is an open question: “We see evidence for increased mTOR activity only in certain tissues of the S47 mouse, so the metabolic impact of this genetic variant appears to be influenced by tissue type and cellular environment. At present we do not know if this tissue specificity is due to differences in GSH level, or to altered mTOR-Rheb or GAPDH-Rheb interactions in different tissues, or to other parameters.”

Reviewer #3:[…] While some of the findings are interesting, it is unclear whether the changes in mTORC1 activity are connected to the observed metabolic phenotypes. Surprisingly, the authors have overlooked the well-known primary modes of regulation of mTORC1 activity (e.g., mTOR/Rheb localization at the lysosome, TSC phosphorylation, and AMPK signaling), but instead focus on the obscure GAPDH-Rheb-mTOR interaction, the contribution of which to mTORC1 activation is unclear and only reported under low-glucose conditions. More work is necessary to substantiate the authors' conclusions.

We agree with the reviewer that the Rheb-mTOR-GAPDH axis was an unusual place to end up, but we now make clearer in the text that we turned to this pathway only when all of our efforts analyzing other conventional mTOR regulators did not yield differences. We now show in supplemental data some of our cumulative data that the WT/S47 variation does not appear to impact p-AMPK or TSC2 levels or Rheb localization at the lysosome (Figure 4—figure supplement 1A, B and E). We have also clarified the text to this regard.

Importantly, we now also confirm in mouse skeletal muscle the major findings of Lee et al., 2009 in Supplemental figure 4- figure 1C, where we show that the Rheb-GAPDH interaction exists in skeletal muscle, and moreover that this interaction is regulated by glucose deprivation (as shown in cultured cells in Lee et al., 2009). We thank the reviewer for this suggestion, as we feel that addressing it has strengthened our findings.

Essential revisions:1) The metabolic phenotypes that the authors associate with mTORC1 signaling are based on correlative observations. The authors should attempt to rescue some of the observed phenotypes in Figure 2D, 2E, and Figure 6 using both allosteric (Rapamycin) and catalytic (Torin-1) inhibitors of mTORC1.

This is an excellent suggestion. As requested, we have now performed the Seahorse experiments with Torin1 (new Figure 3C and D), and found that these data corroborate our findings with rapamycin. With regard to exercise studies, as described above in reviewer 1 point 1, we now clearly state:

“One caveat of this study, however, is that we do not directly demonstrate that the increased mTOR activity in S47 mice is causing their increased lean content or superior performance on treadmill assays. […] It remains to be tested if these are the pathways affected in S47 mice.”

2) Figure 3: What are the authors' thoughts on Rapamycin failing to decrease OCR in S47 cells to the same extent as in WT cells? Is this due to partial inhibition of mTORC1 activity in these cells? Are the FKBP12 levels similar in these cells? Is ECAR dependent on mTORC1? The authors should also use a catalytic inhibitor of mTORC1 (e.g., Torin-1) and compare its effect to the effect of Rapamycin. It would be necessary to determine inhibition of mTORC1 signaling by these inhibitors on multiple substrates of mTORC1 (at least, S6K1 and 4EBP1) to be able to draw conclusions on this.

As requested, we have used Torin1 and found similar effects to rapamycin; also as suggested by this Reviewer, we have performed a time course analysis on WT and S47 LCLs treated with Torin1 and rapamycin, and show that these agents appear comparable in their ability to inhibit markers of mTOR activity in WT and S47 cells (new Figure 3E; new Figure 3—figure supplement1A). We also show that very high concentrations of Torin1 (1 uM) are capable of fully inhibiting OCR in both WT and S47 LCLs (new Figure 3—figure supplement 1B). The combined data suggest that it is the sensitivity of OCR to mTOR inhibition, and not the ability of the drugs to inhibit mTOR, that is distinct between WT and S47 cells. We propose that there may be mTOR-independent effects of S47 on OCR.

3) Figure 5: More work is needed to substantiate the mechanistic conclusions on increased mTORC1 activity in S47 cells because of an enhanced mTORC1-Rheb association and reduced GAPDH-Rheb. Using Proximity Ligation Assay (PLA) assays, the authors suggest that the Rheb-GAPDH association is dependent on the redox status of GAPDH, and in a reduced state as in S47, the GAPDH-Rheb interaction would decrease, allowing for more Rheb to associate with mTOR. While this is an interesting hypothesis, the authors should first validate the reagents used in their studies and employ orthogonal methods to verify these findings. The antibodies used for the PLA should be validated using knockdown experiments followed by PLA. Complementary experiments such as immunoprecipitation assays should be used to test the mTOR-Rheb and GAPDH-Rheb localization interaction changes in WT vs. S47 contexts.

These excellent points are very similar to reviewer 1 Essential revision #1, and we have taken them very seriously. These corroborative approaches have been done. Specifically:

a) We show that GSH impacts mTOR activity, as five hour treatment of cells with the GSH alkylator DEM, and more dramatically the combination of DEM with glutamate (inhibits cystine import), causes a significant decrease in mTOR activity in S47 cells (new Figure 5B).

b) Moreover, by IP-western we now show: exogenous GSH decreases the Rheb-GAPDH complex in WT cells, while conversely, either DEM or BSO (the latter decreases GSH biosynthesis) increases the Rheb-GAPDH complex in S47 cells, as shown by IP-western. These treatments do not affect the steady state level of Rheb or GAPDH. (New Figure 5D). We hope the reviewer agrees that these new data strengthen our conclusions, and again we are grateful for the suggestions.

4) Since the primary mode of activation of mTORC1 is at the lysosomes, the authors should test whether mTORC1, Rheb and TSC2 localization on the lysosomes is affected or not in the S47 cells. No conclusions regarding overactivation of mTORC1 in S47 cells can be made without this crucial bit of data.

This has been done. We now show co-IF experiments for LAMP1-Rheb and Rheb-TSC2 and do not see differences between WT and S47 cells (see new Figure 4—figure supplement 1E).

5) Figure 2F, G and Figure 2—figure supplement 1A, B: The authors conclude a higher flux of glucose carbon into the TCA cycle in S47 cells. Why do the authors think that these minute changes (the differences appear to be less than 5%) are biologically significant? Can the authors observe significant changes in glycolytic intermediates in the 13C6-Glucose tracing experiment? Could the changes in OCR be due to an effect on NAD/NADH instead?

The metabolomics findings were modest, as they were performed after a 15 minute incubation with 13C-glucose. Unfortunately, due to quarantine our colleague was unable to repeat these findings using a longer timepoint in this experiment; therefore, we have moved these findings to Supplemental data. Similarly, we regret that we were unable due to time constraints to assess NAD/NADH levels and agree with this reviewer that this remains an interesting question to address.

[Editors' note: further revisions were suggested prior to acceptance, as described below.]

All reviewers have now reviewed your outline for revision experiments and agree that these proposed experiments will likely go a long way to strengthening the link between p53 S47, GSH, and mTORC1 activity in vitro. […] In this event you acknowledge and agree that the these will be published under the terms of the Creative Commons Attribution license.

We have addressed all of the concerns from the previous review, and the result is a greatly strengthened manuscript.

In sum, our work highlights a novel mechanism whereby the p53 tumor suppressor protein regulates mTOR activity, via the control of cellular redox state. Notably, this work uses as its foundation mouse and human tissues and cells containing an African-centric variant of p53, so the work has direct implications for understanding cancer disparities in African-descent populations. This work suggests that enhanced mTOR activity and metabolic efficiency may have been selected for in early Africa, making this variant one of a growing number of cancer-predisposing variants that possesses other activities that may be under positive selection. All of the authors have seen and approve this work.